# Heterogeneous oxidation of amorphous organic aerosol surrogates by $O_3$, $NO_3$, and OH at typical tropospheric temperatures

Jienan Li, Seanna M. Forrester, Daniel A. Knopf

School of Marine and Atmospheric Sciences, Stony Brook University, Stony Brook, NY 11794-5000, USA

*Correspondence to*: Daniel A. Knopf (Daniel.knopf@stonybrook.edu)

**Abstract.** Typical tropospheric temperatures render possible phase states of amorphous organic aerosol (OA) particles to solid, semi-solid and liquid. This will affect the multiphase oxidation kinetics involving the organic condensed phase and gaseous

oxidants and radicals. To quantify this effect, we determined the reactive uptake coefficients ($\gamma$) of $O_3$, $NO_3$ and OH by substrate films composed of single and binary OA surrogate species under dry conditions for temperatures from 213 to 313 K. A temperature-controlled coated-wall flow reactor coupled to a chemical ionization mass spectrometer was applied to determine $\gamma$ with consideration of gas diffusion transport limitation and gas flow entrance effects, which can impact heterogeneous reaction kinetics. The phase state of the organic substrates was probed via the poke-flow technique, allowing

the estimation of the substrates' glass transition temperatures. $\gamma$ values for $O_3$ and OH uptake to a canola oil substrate, $NO_3$ uptake to a levoglucosan and a levoglucosan/xylitol substrate, and OH uptake to a glucose and glucose/1,2,6-hexanetriol substrate have been determined as a function of temperature. We observed the greatest changes in $\gamma$ with temperature for substrates that experienced the largest changes in viscosity as a result of a solid-to-liquid phase transition. Organic substrates that maintain a semi-solid or solid phase state and as such a relatively higher viscosity, do not display large variations in

heterogeneous reactivity. From 213 K to 293 K, $\gamma$ values of $O_3$ with canola oil, of $NO_3$ with a levoglucosan/xylitol mixture, and of OH with a glucose/1,2,6-hexanetriol mixture and canola oil, increase by about a factor of 34, 3, 2 and 5, respectively, due to a solid-to-liquid phase transition of the substrate. These results demonstrate that the surface and bulk lifetime of the OA surrogate species can significantly increase due to the slowed heterogeneous kinetics when OA species are solid or highly viscous in the middle and upper troposphere. This experimental study will further our understanding of the chemical evolution

of OA particles with subsequent important consequences for source apportionment, air quality, and climate.

## 1 Introduction

Organic aerosol (OA) particles are ubiquitous and can represent 20%-90% of the mass fraction of the submicron aerosol (particles $\leq$ 1 µm in diameter) in the atmosphere. The significance of OA has long been established, influencing air quality, human health, cloud formation process and the radiative budget, on a regional and global scale, and thus climate (Hallquist et

al., 2009;Jimenez et al., 2009;Seinfeld and Pandis, 2016;Stocker et al., 2013;Knopf et al., 2018;Abbatt et al., 2019;Shiraiwa et

al., 2017b;Kanakidou et al., 2005;Pachauri et al., 2014;Pöschl and Shiraiwa, 2015). Characterizing these impacts crucially depends on our ability to determine and quantify aerosol sources and strengths (Bai et al., 2013;Robinson et al., 2006;McFiggans et al., 2019;Hopke, 2016) and to understand the physical and chemical transformation of aerosol particles during atmospheric transport by multiphase chemical processes (George and Abbatt, 2010;Rudich et al., 2007;Laskin et al., 2015;Springmann et al., 2009;Kaiser et al., 2011;Ervens et al., 2011;Zhou et al., 2019;Moise et al., 2015). Gas-to-particle, also termed heterogeneous, reactions can involve organic components in the condensed phase and gas-phase oxidants such as $O_3$, $NO_3$, and OH. These reactions, which can include multiple phases, change the physicochemical properties of particles including composition, density, and hygroscopicity, thereby defining their chemical lifetime, optical properties, and ice nucleating ability (Jimenez et al., 2009;Kroll et al., 2015;Katrib et al., 2005b;Knopf et al., 2018;Slade et al., 2017;Robinson et al., 2007;Shiraiwa et al., 2017a;Moise et al., 2015;Shiraiwa et al., 2012;Murray et al., 2010;Wang et al., 2012).

It is now well established that OA can exhibit amorphous phase states (Mikhailov et al., 2009;Virtanen et al., 2010;Koop et al., 2011;Reid et al., 2018;Renbaum-Wolff et al., 2013;Kidd et al., 2014). Depending on the viscosity and microstructure, the amorphous phases can be classified as glasses, rubbers, gels, or ultra-viscous liquids (Mikhailov et al., 2009;Riemer et al., 2019). The glass transition temperature, $T_g$, is a characteristic parameter to describe the viscosity of a liquid on the order of $10^{12}$ Pa s (Koop et al., 2011;Angell, 1995;Zobrist et al., 2011;Dette and Koop, 2015;Zhang et al., 2019). At this temperature, the molecular motion of the species is so slow that it can be considered a solid. The phase state of OA can be modulated by particle composition and environmental conditions such as relative humidity (RH) and temperature (Koop et al., 2011;Zobrist et al., 2008;Shiraiwa et al., 2017a;Petters et al., 2019). Particle viscosity will influence the diffusion of atmospheric oxidants and other small gas molecules (e.g., $O_3$, OH, $NO_3$, $H_2O$) entering the organic matrix (Price et al., 2015;Zobrist et al., 2011;Slade et al., 2017;Davies and Wilson, 2016;Moridnejad and Preston, 2016) as well as the transport and mixing of the condensed-phase organic species (Rothfuss and Petters, 2017;Marsh et al., 2018;Lienhard et al., 2015;Abramson et al., 2013;Chenyakin et al., 2017;Kiland et al., 2019;Shiraiwa and Seinfeld, 2012;Renbaum-Wolff et al., 2013). Therefore, it can be expected that the particle phase state will significantly affect the rate of multiphase chemical kinetics (Slade and Knopf, 2014;Kolesar et al., 2014;Knopf et al., 2005;Katrib et al., 2005a;Pöschl and Shiraiwa, 2015;Davies and Wilson, 2015;Gaston et al., 2014;Zhang et al., 2018;Houle et al., 2018b). The heterogeneous reaction kinetics are often expressed by the reactive uptake coefficient (γ), which represents the fraction of gas collisions with a substrate surface that yield uptake or reaction (Pöschl et al., 2007;Schwartz, 1986).

As has been demonstrated for heterogeneous ozonolysis reactions, the composition of the condensed phase can alter the phase state, thereby, significantly altering the reactive uptake kinetics (Knopf et al., 2005;Hearn et al., 2005;de Gouw and Lovejoy, 1998;Ziemann, 2005). In general, the solid condensed phase showed significantly lower $\gamma_{O_3}$ values compared to the ozonolysis of the liquid phase. Recent experimental studies focused on how relative humidity (RH) influences the reactive uptake of gas oxidants via its impact on the phase state of the organic species (Berkemeier et al., 2016;Steimer et al., 2015;Shiraiwa et al., 2011;Slade and Knopf, 2014;Li et al., 2018;Davies and Wilson, 2015;Hu et al., 2016;Marshall et al., 2018;Pajunoja et al., 2016). The heterogeneous oxidation of a typical component of biomass burning aerosol (BBA) particles,

levoglucosan (LEV) by OH yielded γ values in the range from 0.008 to 1 with implications for the lifetime of LEV ranging from weeks at dry conditions (Slade and Knopf, 2013;Kessler et al., 2010;Slade and Knopf, 2014;Arangio et al., 2015) to a couple of days when LEV is in a more liquid-like state in response to ambient RH (Yang et al., 2013;Bai et al., 2013;Slade and Knopf, 2014). The general conclusion is that at lower RH, the reactive uptake can be dominated by surface reactions when the condensed phase is in a solid state, whereas at higher RH, the condensed phase can be semi-solid or liquid and the oxidation

process can commence at the surface and in the bulk. For the latter scenario, the gaseous oxidants can access greater depths of the condensed phase, and the oxidized organic species at the surface can be readily replenished by unreacted molecules from the bulk (Arangio et al., 2015;Slade and Knopf, 2014). Davies and Wilson (2015) investigated how the viscosity change of citric acid, due to changes in water content, governs the reactive uptake of OH radicals. They observed that the depletion of citric acid and the formation of reaction products are confined near the aerosol-gas interface on the order of 8 nm at 20% RH,

and the reaction depth increases to ~50 nm at 50% RH. Recently, Li et al. (2018) observed that the heterogeneous aging of SOA by OH radicals at 89% RH and 25% RH resulted in ~60% and 20% loss of particle mass, respectively. The authors concluded that this difference in particle mass degradation is attributed to a larger OH uptake coefficient and/or larger fragmentation probability at higher RH.

The temperature in the troposphere ranges approximately from 200 K to 300 K (Wallace and Hobbs, 2006), with

subsequent impact on the OA phase state (Shiraiwa et al., 2017a;Koop et al., 2011) and multiphase reaction kinetics. However, only a few laboratory studies investigated the temperature dependence of heterogeneous oxidation reactions while considering the phase transition of organic substrates and aerosol particles (de Gouw and Lovejoy, 1998;Edebeli et al., 2019;Gross et al., 2009;Knopf et al., 2005;Moise and Rudich, 2002;Slade et al., 2017;Moise and Rudich, 2000;Moise et al., 2002). Moise and Rudich (2002) demonstrated that the uptake of ozone by oleic and linoleic acid is a strong function of temperature, where γ

decreases by about an order of magnitude with decreasing temperature when both substrates transformed from a liquid to a solid state. Similarly, Gross et al. (2009) observed a decrease of 70% to 90% for γ of $NO_3$, as oleic acid, diethyl sebacate, and conjugated linoleic acid substrates solidified as the temperature decreased from 302 K to 263 K. The authors suggested that the net liquid phase reaction involves both a surface reaction and a bulk reaction, whereas solidification of the substrate greatly minimizes the significance of any bulk reactions. This is in line with results of a kinetic flux model applied to $NO_3$ exposure

studies of solid LEV and abietic acid substrates, further indicating that under shorter time scales reactive uptake was dominated by surface reaction, whereas for times scales greater than ~100 s, even for a solid organic substrate, bulk processes can impact the overall reactive uptake kinetics (Shiraiwa et al., 2012) Very recently, Edebeli et al. (2019) investigated the temperature dependence of bromide oxidation by ozone in a citric acid/bromide mixture and observed that γ decreases from $2 \times 10^{-6}$ at 289 K to $0.5 \times 10^{-6}$ at 245 K. Their analysis indicated that the humidity-driven acceleration in uptake reactivity decreased due to

the increased viscosities of the citric acid/bromide mixture and decreased diffusivity of ozone as the temperature was lowered. The effect of temperature on uptake kinetics via its impact on the condensed phase state has also been highlighted in a recent modeling study (Mu et al., 2018), which showed that low temperature conditions can substantially increase the lifetime of PAHs against $O_3$ and enhance their dispersion through both the planetary boundary layer and the free troposphere. Temperature

also impacts the other processes involved in multiphase kinetics (Schwartz, 1986;Pöschl et al., 2007) including the collision flux of gas-phase species, the desorption rate and surface and bulk reaction rates, both typically expressed by an Arrhenius factor (Laidler et al., 1940;Pöschl et al., 2007;Baetzold and Somorjai, 1976). Clearly, the temperature dependency of the reaction kinetics and the role of the condensed phase state pose a challenge for detangling the underlying physicochemical processes governing multiphase chemical kinetics of OA under typical tropospheric conditions.

To add to our understanding of multiphase chemical kinetics at low temperatures, we determined the reactive uptake of $O_3$, $NO_3$ and OH by OA surrogates for temperatures ranging from 213 K to 293 K under dry conditions. We employed a chemical ionization mass spectrometer coupled to a coated-wall flow-tube reactor. As OA proxies, we applied substrates of canola oil (CA), levoglucosan (LEV), a levoglucosan/xylitol mixture (LEV/XYL), glucose (GLU), and a glucose/1,2,6-hexanetriol mixture (GLU/HEX). CA is a mixture of multiple saturated and unsaturated fatty acids, however, dominated by unsaturated oleic and linoleic acid, and serves as OA surrogate for both marine and terrestrial organic aerosol including anthropogenic emissions like cooking (Kawamura et al., 2003;Schauer et al., 2002;Limbeck and Puxbaum, 1999;Liu et al., 2017;Rogge et al., 1991). LEV serves as a surrogate for biomass burning aerosol (BBA) including natural sources such as forest fires and anthropogenic sources such as heating and cooking BBA (Schauer et al., 2001;Iinuma et al., 2007). GLU can serve as a tracer to characterize and apportion primary biogenic organic aerosols (PBOAs) (Zhu et al., 2015;Samaké et al., 2019). XYL and HEX are used in compounds mixtures to modify the glass transition temperature, thus, condensed-phase viscosity. $O_3$ and OH radicals reflect typical gas oxidants present in photochemically active regions and during long-range transport. $NO_3$ radicals reflect an effective oxidant participating in the nighttime chemistry of typically polluted regions (Finlayson-Pitts and Pitts Jr, 1999;Brown et al., 2006). Some of the examined substrates undergo a phase transition from liquid to solid in the probed temperature regime with subsequent consequences for the heterogeneous uptake kinetics. The substrate's phase state was qualitatively verified by conducting film poke-flow experiments. The results of our reactive uptake experiments emphasize the importance of the temperature-induced phase changes of the OA surrogates when describing the chemical degradation of OA during atmospheric transport.

## 2 Experimental methods

### 2.1 Apparatus

The experimental system is based on our previous setups (Slade and Knopf, 2013;Knopf et al., 2011;Knopf et al., 2005) and includes a temperature-controlled coated-wall flow reactor and a custom-built chemical ionization mass spectrometer (CIMS), as shown in Fig. 1. It consists of three parts: gas-phase oxidant generation, multiphase chemical reaction, and oxidant detection. The gas-phase oxidant ($O_3$, $NO_3$, or OH) enters the coated-wall flow reactor via a movable injector, where it can react with the organic substrate film coating on the inner wall of a rotating tube. The changes in oxidant concentration are measured by CIMS via soft ionization in the chemical ionization region.

The uptake measurements are conducted in a temperature range of 213 to 313 K. The axial temperature variation in the flow reactor is smaller than 1 K for all our experimental conditions, which is determined with a thermocouple that is fixed at the tip of the injector, as discussed in the Supplement S1. The rotating tube fits snuggly inside the flow tube that is enclosed by a cooling jacket for temperature control. The rotating speed of the substrate tube is set to ~5-10 rpm to keep the liquid film evenly distributed on its inner surface. The pressure in flow reactor ranges from 2 to 5 hPa depending on the applied oxidant

gases.

For OH radical uptake experiments, an additional rotatory pump is directly connected to the flow reactor. The pump is used to lower the flow reactor pressure to < 2.5 hPa, thus minimizing gas transport limitation due to diffusion (Knopf et al., 2015;Zasypkin et al., 1997) while simultaneously maintaining a high flow rate. Two sizes of rotating glass cylinders with diameters of 1.75 cm and 1.2 cm are used in these uptake experiments. The latter one is designed specifically for OH

heterogeneous reactions to avoid limitation of gas transport by diffusion. For $O_3$ and $NO_3$ uptake experiments, gas transport limitations due to diffusion can be neglected under applied flow conditions because of a slower reactive uptake.

The pseudo-first-order loss of the oxidant species to the organic substrate is determined by monitoring the loss of the gaseous oxidants as the injector is pulled back in 1 or 2 cm increments until reaching 10 cm. The flow rate of gas oxidants in the movable injector ranges from 2 to 100 $cm^3 min^{-1}$ STP (standard temperature and pressure). The flow rate of the carrier gas

He in the flow reactor ranges from 30 to 900 $cm^3 min^{-1}$ STP. As a result, the residence time varies two orders of magnitude in these experiments, ranging from ~2 ms for OH uptake to 100 ms for $O_3$ uptake. The Reynolds number (Re) for all experiments is below 20, indicating laminar flow conditions. Care has been taken to ensure a carrier-to-injector gas velocity ratio of >1.33, in order to avoid the disruption of the gas flow by a fast gas flow exiting the injector, as pointed out by Davis (2008). This condition is also a necessity to accurately account for the gas flow entrance effect, which is important for fast uptake kinetics,

as further discussed in the Appendix. The γ value at each temperature reported in this study is derived from at least 6 reactive uptakes, with a freshly prepared organic substrate employed for each.

## 2.2 Oxidant formation, detection and flow conditions

A carbon filter (Supelco, Superlcarb HC) and a Drierite cold trap cooled with liquid nitrogen or an ethanol/dry ice mixture are used to purify ultra-high purity (UHP) gases of $N_2$, He and $O_2$. $O_3$ is generated by introducing a flow of $O_2$ through a UV

source (Jelight, model #600) prior to the injector. The flow rate of the $O_2/O_3$ mixture ranged between 2 - 7 $cm^3 min^{-1}$ (STP). A carrier He gas flow of 30-85 $cm^3 min^{-1}$ (STP) enters the flow reactor, mixes with the injector $O_2/O_3$ gas flow, and results in a laminar flow with Re < 3. $O_3$ is detected as $O_3^-$ after charge transfer reaction with $SF_6^-$. $SF_6^-$ is generated by passing a minor amount of $SF_6$ in $N_2(g)$ through a $^{210}Po$ source. The $O_3$ concentrations ranged between $1.5\times 10^{11}$ - $2.1 \times 10^{12}$ molecules $cm^{-3}$, reflecting typical atmospheric concentrations (Wallace and Hobbs, 2006;Finlayson-Pitts and Pitts Jr, 1999).

$NO_3$ is generated by thermal dissociation of $N_2O_5$ gas at 433 K in a glass oven prior to the injector, as previously (Knopf et al., 2006;Knopf et al., 2011). $N_2O_5$ is generated via reaction of $NO_2$ and $O_3$ and stored as pure $N_2O_5$ crystals in a glass container at 193 K. A gas flow of $N_2O_5$ in He of about 12-18 $cm^3 min^{-1}$ STP is further diluted by 100-120 $cm^3 min^{-1}$ (STP)

He(g) flow before entering the injector. The flow reactor carrier gas is composed of He and $O_2$ at ~500 and ~100 $cm^3\,min^{-1}$ (STP), respectively. This results in laminar flow conditions (Re < 10). $NO_3$ is detected as $NO_3^-$ after chemical ionization by $I^-$.

$NO_3$ concentrations are obtained via titration of a known amount of NO, resulting in $8 \times 10^9$ to $7 \times 10^{11}$ molecules $cm^{-3}$, representing typical atmospheric $NO_3$ nighttime concentrations (Finlayson-Pitts and Pitts Jr, 1999;Brown et al., 2006).

OH radicals are produced following previously established methods (Bertram et al., 2001;Slade and Knopf, 2014) via the reaction of H atoms with $O_2$. A 2.45 GHz Evenson microwave cavity powered by a microwave generator is utilized to maintain an H plasma. The H atoms are formed by flowing a mixture of $H_2(g)$ (1-2 $cm^3\,min^{-1}$, STP) and He(g) (40-80 $cm^3\,min^{-1}$, STP)

through a 1/4" OD (outer diameter). Pyrex tube that runs through the center of the microwave cavity. The generated H atoms then pass through a 1/8" OD. Pyrex tube directed down the center of a 1/4" OD Pyrex injector where OH production occurs with H radicals reacting with $O_2(g)$ at 2-22 $cm^3\,min^{-1}$ (STP). Complete reaction to OH is ensured by monitoring the formation of $OH^-$ via charge transfer with $SF_6^-$ in the CIMS. At colder flow reactor temperatures, the usable injector length is shortened to avoid artifacts in OH concentration due to temperature gradients in the injector tube which can impact OH formation

(Atkinson et al., 2004). A He gas flow of 700-900 $cm^3\,min^{-1}$ (STP) enters the flow reactor and further mixes with the injector flow, yielding a laminar gas flow with Re < 20. Employed OH concentrations ranged between $5\times10^7$ and $1\times10^9$ molecule $cm^{-3}$, which are higher than typical background OH concentrations of $10^6$ molecule $cm^{-3}$ (Finlayson-Pitts and Pitts Jr, 1999).

### 2.3 Substrate film preparation and characterization

The film substrate preparation method when using a coated-wall flow reactor has been discussed in detail in previous studies

(Knopf et al., 2011;Slade and Knopf, 2013). Canola oil is liquid and thus can be directly applied on the inner surface of the rotating tube. The other organic substrates, i.e., LEV, LEV/XYL mixture, GLU and GLU/HEX mixture, are first dissolved in water and then 1-2 ml of the aqueous solution is evenly distributed inside the rotating glass tube. The concentration of the LEV solution is 5% (w/w) due to its relatively lower solubility, and all other solutions are 10% (w/w). The organic mass ratios of the LEV/XYL and GLU/HEX films are 1:1 and 4:1, respectively. As outlined further below, these mass ratios were chosen to

set the expected glass transition temperature within the examined temperature range. During the entire process, a dry He or $N_2$ purging gas flow is applied to avoid contamination by room air. The glass tube is rotating in the flow reactor until a smooth coating is established. To dry the substrate, the substrate is exposed to rough vacuum conditions (1~2 hPa) resulting in the evaporation of water. Surface solidification of single-component saccharide films may result in trapping of condensed-phase water and thus impacting the film composition and its viscosity. The remaining water content for all substrate films after the

drying process were determined to be 13%-16% (w/w) by measuring the mass change of the organic substrate films before and after the drying process with an ultra-microbalance (Mettler Toledo XP2U), as detailed in Supplement S2. The thickness of the applied organic film is estimated to be around 50-100 µm. We assume a smooth surface morphology and use the geometric surface area of the film to derive γ.

The phase state of the applied organic substrates at different temperatures can be qualitatively verified by using the poking

experiments outlined below. The "poke-flow" technique has been applied to quantitatively estimate the viscosity of amorphous

organic particles (Murray et al., 2012) including secondary organic material (SOM) produced by α-pinene ozonolysis (Renbaum-Wolff et al., 2013). A similar setup is used here to qualitatively investigate the phase state of applied amorphous substrates at different temperatures ranging from 213 to 313 K. A temperature-controlled cooling stage (Linkam BSC 196) is coupled to an optical microscope (Olympus BX51) (Fig. 2). Images of the film substrate are recorded at 230 magnification and the depth of field of the 10X objective is 15.9 µm. The organic film substrate, generated from a volume of 2 µl liquid applied to a glass slide, is first placed into the flow reactor, undergoing the same preparation method as the substrate films employed for reactive uptake experiments. This results in a film with a thickness of about 50-100 µm. The substrate film is then moved into the poke-flow experiment. In the cooling stage, the organic substrate film is sealed against room air and resides in an atmosphere of dry $N_2$ at positive pressure during the entire poking process. After poking with a needle, the phase states and flow characteristics of the organic substrate films at different temperatures are determined under the microscope as a function of time. Images are recorded with the film surface in focus to monitor the flow of the organic substrate and to probe film viscosity. Under the applied resolution, all film substrates appeared to be smooth and transparent. The calibration of the cooling stage was performed via measurements of the melting points of octane (216.35 K), decane (243.5 K) and water (273.15 K). Before each experiment, the temperature was verified by measuring the melting temperature of three 2 µL water droplets (273.15 K). The difference to the expected melting temperature difference is always smaller than 1 °C. We measured $T_g$ with at least five independently prepared substrate films.

**2.4 Chemicals**

Listed below are the chemicals we used in our study, and corresponding purities and manufactures. $N_2$ (ultra-high purity, UHP), $O_2$ (UHP), $H_2$ (UHP) and He (UHP) were purchased from Airgas East. $SF_6$ (99.998%) was acquired from Praxair. $NO_2$ (99.5%) was purchased from Matheson. 1,6-anhydro-b-D-glucopyranose (99%) was purchased from Acros Organics. Glucose (99%) and xylitol (99%) were acquired from Alfa Aesar. 1,2,6-Hexanetriol (96%) and iodomethane (99%) were purchased from Sigma-Aldrich. Purity of canola oil was not determined. Millipore water (resistivity >18.2 MΩ cm) was applied to prepare aqueous solutions for generation of organic substrates.

**3 Results and Discussion**

**3.1 Characterization of substrate phase state**

Images of the film surfaces in Fig. 3 show the flow characteristics and the morphologies of a GLU/HEX substrate mixture when subject to poking by a needle. In the experiment at $T = 293$ K (Fig. 3A), the film surface is deformed after being poked, leaving a black dent. The area around the dent reflects the smooth morphology of the film substrate before poking it. The slightly darker shades within the imaged area do not represent any morphology features but the unfocused scratches from the silver substrate holder beneath the film. The film flowed at an observable rate to restore a smooth surface, minimizing the

surface energy of the system (Fig. 3A). At 273 K, the same deformed film did not show observable changes within 30 minutes (Fig. 3B). Only after 8 hours, minor changes in film morphology can be identified, clearly demonstrating greater substrate viscosity at this temperature compared to 293 K. At 253 K, the notch by poking on the surface maintains its shape for 8 hours, indicative of a semi-solid phase state (Fig. 3C). At $T = 233$ K, which is below the predicted $T_g = 248$ K, the surface shattered when poked with a needle (Fig. 3D). Furthermore, over the experimental observation time of 8 h, no restorative flow was observed at 233 K. The fragments have clear glass-like cracks, and no smoothing of the edges was observed over the course of the experiment. Based on our poke-flow experiments as shown in Figure S2-S7, the applied GLU and LEV substrates remain in a solid or semi-solid phase state within the temperature range of 213 K to 313 K. However, the mixtures of GLU/HEX and LEV/XYL exist as liquids at 293 K and experience a glass phase transition with decreasing temperature.

By monitoring the substrate surface morphology, we can estimate $T_g$ of the applied substrates as the highest temperature when shattering occurs. Viscosity greater than $10^{12}$ Pa s indicates the presence of a glassy phase. Thus, we ascribe the observed shattering of the substrate film a viscosity of $10^{12}$ Pa s, although the exact value cannot be assessed with the poke-flow technique. All images of the poke-flow experiment are documented in Figs. S2-S7 in the Supplement S3. Table 1 gives the estimated $T_g^{\text{exp}}$ values of applied substrates compared to literature $T_g^{\text{lit}}$ and predicted $T_g^{\text{pred}}$ values. $T_g^{\text{pred}}$ for substrate mixtures is derived from the Gordon Taylor equation (Gordon and Taylor, 2007) assuming literature $T_g$ values and no residual water present. The uncertainties in $T_g^{\text{pred}}$ values are derived by Gaussian error propagation. Except for the GLU substrate film, we achieve agreement or close agreement with either $T_g^{\text{lit}}$ or $T_g^{\text{pred}}$. Since $T_g$ determination depends on cooling or drying rates, we do not expect agreement with literature values or predicted values. In addition, if 10%-16% residual water is considered while using Gordon-Taylor equation, $T_g^{\text{pred}}$ would be roughly 30 K lower than $T_g^{\text{exp}}$, assuming the residual water is homogeneously distributed in the film. This discrepancy is very likely due to our substrate film preparation process where under slow drying (evaporation), the outermost layers of the film contain less water than the deeper layers. Thus, the substrate surface represents closer the experimental conditions. Furthermore, the microscope focus in our poke-flow experiments is on the substrate surface, thereby monitoring the substrate morphology that is governed by the film viscosity closest representing the desired conditions.

The temperature dependence of the substrate viscosity can be predicted by using the modified Vogel-Tammann-Fulcher (VTF) equation (Angell, 1991):

$$\log \eta = -5 + 0.434 \frac{T_0 D}{T - T_0} , \tag{1}$$

where $T_0$ is the Vogel temperature and $T$ is the ambient temperature. The fragility parameter, $D$, is defined in terms of the deviation of the temperature dependence of the viscosity from the simple Arrhenius behavior. Assuming $T = T_g$, $\eta = 10^{12}$ Pa s, $T_0$ can be represented by $T_g$ (DeRieux et al., 2018):

$$T_0 = \frac{39.17 T_g}{D + 39.17} . \tag{2}$$

Therefore, the temperature dependence of substrate viscosity can be predicted using our measured $T_g^{exp}$ as illustrated in Fig. 4. For our calculations, we use literature $D$ values for pure sugars and a mass-weighted interpolation of $D$ values for the mixtures as shown in Table 1. In Fig. 4a, LEV maintains a relatively higher viscosity compared to the LEV/XYL mixture throughout the studied temperature. The poke-flow experiments indicate the presence of a liquid phase at 293 K and solid phase at 238 K. In contrast to our poke-flow experiments, Fig. 4a suggests that the LEV/XYL mixture does not show a liquid phase in the examined temperature regime. Capturing this phase transition with the modified VTF equation (Angell, 1991), can be only achieved when using lower $D$ values of 5.7~7.6 (compared to the interpolated $D$ value of the LEV/XYL mixture), a reasonable assumption when considering the presence of residual water in the substrate. The addition of water increases the steepness of the glass transition and decreases $D$ values by potentially reducing the thermal energy necessary to promote the cooperative chain motions (Angell, 2002;Borde et al., 2002). Furthermore, a study of the trehalose/water system corroborates that the presence of water lowers $D$ values to 3.32 - 4.85 as a lower limit (Elias and Elias, 1999). For GLU and the GLU/HEX mixture shown in Fig. 4b, adding HEX to GLU greatly lowers the $T_g$ of the mixture, and thus decreases its viscosity. Also, in this case, the predicted substrate viscosity of GLU/HEX disagrees with our observations when interpolating the $D$ values of the pure compounds. A $D$ value between 4.3 and 5.7 better represents the observed phase transition of the GLU/HEX mixture.

**3.2 Reactive uptake kinetics**

The uptake coefficient is determined experimentally from the loss of the gas phase oxidant to the organic substrate as the substrate area or corresponding reaction time is changed. Figure 5 shows three exemplary uptakes of $O_3$ by the canola oil (CA) film, $NO_3$ by LEV film and OH radical by the GLU film, indicating the stepwise irreversible removal of the gas oxidants. Taking $O_3$ uptake as an example, the injector is pulled back in 2 cm increments until reaching 10 cm and is then pushed back to its original position (0 cm), with normalized $O_3$ signals recovering to unity.

From these data, the observed first-order wall loss rate, $k_{obs}$, is determined as the slope of the change in the natural logarithm of the oxidant signal as a function of reaction time, i.e., the residence time of the oxidants in the flow reactor (Knopf et al., 2011;Slade and Knopf, 2013). Figure 6 shows exemplary $k_{obs}$ derived for uptakes of $O_3$, $NO_3$, and OH for various temperatures. $k_{obs}$ is obtained from the slope of the linear fit to the data. The good linear regression indicates that the reported uncertainties for γ are likely due to the variability of organic substrates and uncertainty in the diffusion coefficient. In the case of $O_3$ uptake (Fig. 6a), we can identify two different regimes of $k_{obs}$, where the linear regression with a larger slope at higher temperatures indicates greater reactivity. As discussed in detail further below, the large difference in $k_{obs}$ between these two temperature regimes coincides with the canola oil substrates being in a liquid and solid phase state, respectively. For the cases of $NO_3$ and OH (Figs. 6b and 6c), the gradual change of $k_{obs}$ coincides with a phase transition for both substrates from a highly viscous liquid to a semi-solid and solid (glassy) phase.

The change in concentration of oxidant X along the flow tube due to reactive uptake can be expressed using the effective uptake coefficient

$$\gamma_{\text{eff,X}} = \frac{D_{\text{tube}}}{\omega_X} \times \left[ ln\left(\frac{[X]_{g,0}}{[X]_g}\right)/t \right] = \frac{D_{\text{tube}}}{\omega_X} \times k_{\text{obs}} \quad . \tag{3}$$

Following the Knopf-Pöschl-Shiraiwa (KPS) method (Knopf et al., 2015), actual $\gamma$ can be derived as:

$$\gamma = \frac{\gamma_{\text{eff,X}}}{1 - \gamma_{\text{eff,X}}\frac{3}{2N_{\text{shw}}^{\text{eff}}Kn_X}} \quad , \tag{4}$$

where $D_{\text{tube}}$ is the diameter of the coated-wall tube and $\omega_X$ (X = O$_3$, NO$_3$ or OH radical) is the mean molecular velocity of the respective gas-phase oxidant. $N_{\text{shw}}^{\text{eff}}$ is the effective Sherwood number which represents an effective dimensionless mass-transfer coefficient to account for changes in the radial concentration profile of the entry region (Knopf et al., 2015;Davis, 2008), and $Kn_X$ is the Knudsen number that characterizes the flow regime (Wutz, 1989;Fuchs and Sutugin, 1971). The Sherwood number departs from $N_{\text{Shw}}^{\text{eff}}$=3.66 under conditions of fast flows and short tubes. The KPS method is advantageous over the correction approach by Brown (1978) since it accounts for entrance effects that can result in the overestimation of reactive uptake coefficients as outlined by Davis (2008) and discussed in more detail in the Appendix. The Cooney-Kim-Davis (CKD) method (Cooney et al. 1974) also accounts for the flow entrance effects as discussed in Murphy and Fahey (1987). This is crucial for fast uptakes involving the OH radical, however, less so for the uptakes involving O$_3$ and NO$_3$ radicals. The uncertainty in derived $\gamma$ values represents the greater value of either a 20% uncertainty in the diffusion coefficient or variation in measurements expressed as $\pm 1\sigma$.

The gas-phase diffusion coefficient of O$_3$ in He ($D_{O_3-He}$) is taken as 394 Torr cm$^2$ s$^{-1}$ at 298 K (Moise and Rudich, 2000). Diffusion coefficients of NO$_3$ in He and O$_2$ are taken as 345 and 80 Torr cm$^2$ s$^{-1}$ at 273 K, respectively (Rudich et al., 1996). The method introduced by Fuller et al. (1966) was utilized for O$_3$ and NO$_3$ to derive the diffusion coefficients at other temperatures. The diffusion coefficients of OH in He and O$_2$ gas ($D_{OH-He}$ and $D_{OH-O_2}$) for different temperatures are theoretically calculated based on a method reported by Mason and Monchick (1962). A reference value of $D_{OH-He}$ measured at room temperature is $662 \pm 33$ Torr cm$^2$ s$^{-1}$ (Ivanov et al., 2007;Liu et al., 2009). To calculate the diffusion coefficient of an oxidant ($D_X$) in a mixture of He and O$_2$, we applied the following equation with $P_{He}$ and $P_{O_2}$ representing pressures derived from experimental flow rates and pressure measurements (Hanson and Ravishankara, 1991)

$$D_X = \left(\frac{P_{He}}{D_{X-He}} + \frac{P_{O_2}}{D_{X-O_2}}\right)^{-1} \quad . \tag{5}$$

The pressure in the flow reactor is maintained at less than 2.5 hPa to avoid transport limitations by diffusion of OH radicals that undergo fast reactive uptake kinetics. Diffusion limitation can be estimated by using the additivity formula for kinetic resistances as (Gershenzon et al., 1995)

$$\frac{1}{k_{\text{obs}}} = \frac{1}{k_k} + \frac{1}{k_d} \tag{6}$$

$$k_k = \frac{\omega_X \times \gamma}{R \times (2-\gamma)} \tag{7}$$

$$k_d = 3.66 \frac{D_X}{R^2} \quad , \tag{8}$$

where $k_{obs}$ (s$^{-1}$) is the observed wall loss rate of heterogeneous uptake, and $k_k$ and $k_d$ represent its kinetic and diffusion limits, respectively. $R$ is the radius of the flow tube. Gershenzon et al. (1995) proposed that $k_k/k_d$ should be < 3~5 to obtain an accurate $\gamma$ value. Some of the highest temperature OH uptake experiments involving liquid canola oil were conducted under diffusion limitation with $k_k/k_d \cong$ 15~30. However, for all the remaining experiments $k_k/k_d < 5$ and, thus, are not diffusion limited.

As outlined in Supplemental Text S4, for semi-solid and solid substrate films, the fraction of the unoxidized reaction sites is always larger than 90% due to short reaction time and low concentrations of gas oxidants (Bertram et al., 2001) implying no significant surface saturation effect on $\gamma$. In other words, during the typical duration of a reactive uptake experiment, the oxidant exposure does not lead to complete oxidation of the substrate surface.

## 3.3 Temperature modulated reactive O$_3$ uptake

Figure 7 shows $\gamma$ of O$_3$ reacting with canola oil as a function of temperature. The reactive uptake is determined under dry conditions in the presence of O$_2$. Both Brown and KPS methods were used to derive $\gamma$ (Brown, 1978;Knopf et al., 2015). Both methods yield the same results due to the slow uptake kinetics as further discussed in the Appendix. Around 90% of canola oil is made up of oleic acid, linoleic acid and α-linolenic acid (Ghazani and Marangoni, 2013), and it solidifies readily around its melting point (de Gouw and Lovejoy, 1998). $\gamma$ decreases sharply by a factor of ~18 from $(5.65 \pm 0.89) \times 10^{-4}$ to $(3.23 \pm 0.74) \times 10^{-5}$ as the temperature decreases from 267 K to 258 K, a temperature range in which the canola oil experiences a phase transition from liquid to solid (Fasina et al., 2008). In this temperature regime, the bulk diffusion coefficients of both ozone and canola oil are expected to decrease by several orders of magnitude due to the transition from a viscous liquid to a solid (Koop et al., 2011). Our O$_3$ reactive uptake coefficients agree with previous literature data by de Gouw and Lovejoy (1998) and extend those to lower temperatures. A study by Berkemeier et al. (2016) showed that the uptake of O$_3$ by shikimic acid is ~5×10$^{-6}$ at low RH conditions and increases by a factor of ~16 at higher RH due to a decrease in viscosity as a result of a RH-induced phase transition, similar to our observations. Our results are consistent with previous studies (Shiraiwa et al., 2009;Shiraiwa et al., 2010;Steimer et al., 2015), which showed that the O$_3$ uptake is slower and restricted to near the particle surface if the organic is in a solid phase, while the uptake is dominated by bulk reaction if the organic is in a liquid phase.

In the low temperature regime, from 213 K to 258 K, $\gamma$ only increases by about a factor of 1.7 (Fig. 6). If we consider the gas-phase reaction kinetics of O$_3$ with an unsaturated bond having an activation energy of 15 kJ mol$^{-1}$ (Atkinson et al., 2006), an increase of the reaction rate by a factor of 4.4 would be expected over a temperature of 45 K. This less than expected temperature dependency of $\gamma$ may be due to the combined temperature effects on both the underlying reaction kinetics and the desorption lifetime (Pöschl et al., 2007). As the temperature decreases and reactivity decreases, the desorption lifetime increases, potentially resulting in a compensating effect on the overall uptake kinetics. In the temperature regime above the phase transition, between 267 K and 293 K, $\gamma$ is significantly larger than when the canola oil substrate is solid, likely due to greater reaction rates and greater O$_3$ and condensed-phase diffusion coefficients.

## 3.4 Temperature modulated reactive NO₃ uptake

The uptake coefficient of $NO_3$ by levoglucosan (LEV) decreases from $(4.2\pm0.6) \times 10^{-4}$ to $(2.8 \pm 0.3) \times 10^{-4}$ as the temperature decreases from 293 K to 213 K, thus, displaying only a slight change in γ within the experimental temperature range (Fig. 8a). Based on our poke-flow experiment, LEV substrates maintained a semi-solid or solid phase state for all examined temperatures. The slight increase of γ with temperature may be due to the combined effects of changes in substrate viscosity (see Fig. 4), the counteracting effect of the temperature dependence of the reaction rate (Atkinson et al., 2006) and the desorption lifetime (Pöschl et al., 2007). Assessment of these various impacts on the observed uptake kinetics necessitates an in-depth analysis, e.g., by application of kinetic flux modeling (Shiraiwa et al., 2012;Arangio et al., 2015). Note, the γ of $NO_3$ by LEV in this experiment is smaller than the value of $(1.3 \pm 0.9) \times 10^{-3}$ determined in our previous study at 298 K, although the data agree within uncertainties (Knopf et al., 2011). A possible explanation for the lower γ value in this study is that the substrates have undergone longer drying to remove residual water. Presence of water would render the film less viscous thereby increasing reactivity.

For the mixture of LEV/XYL (Fig. 8b), γ at 293 K is $8.5 \times 10^{-4}$, about a factor of 2 larger than for pure LEV (see Table 2). As the temperature decreases, γ reaches a value of $3.1 \times 10^{-4}$ at 213 K, similar to the γ value derived for solid LEV (Fig. 8a). Our poke-flow experiment yielded $T_g = 238$ K for the applied LEV/XYL film. The largest change of γ is observed above the expected $T_g$ (Fig. 8b). Therefore, the significant change in $NO_3$ uptake reactivity in the investigated temperature range can be attributed to the transition of a solid or highly viscous to a liquid substrate film. As such, the underlying reaction mechanism changes, i.e., at higher temperatures, the kinetic uptake is likely governed by surface and bulk reactions and at lower temperatures only surface reaction dominates. As shown in Shiraiwa et al. (2012), for highly viscous/solid LEV films, the transport of bulk LEV towards the interface is strongly limited, slowing the reactive uptake kinetics. At higher temperatures, with decreasing viscosity of LEV, the diffusivity of LEV and $NO_3$ increases, likely facilitating transport and thus reaction.

Gross et al. (2009) reported that γ of $NO_3$ by liquid glycerol (GLY) is $(14\pm3) \times 10^{-4}$ at 293 K and decreases to $8.3\times10^{-4}$ at 268 K, representing a decrease of 41% in γ over 25 K. The $T_g$ of glycerol is 193 K, and the viscosity decreases from 20 Pa s at 268 K to 1.32 Pa s at 293 K (Schröter and Donth, 2000). For the same temperature difference of 25 K, γ for the viscous liquid LEV/XYL substrate decreases by about 24%. The viscosity of LEV/XYL film is around $10^1$ to $10^2$ Pa s based on our poke-flow experiments and viscosity prediction as outlined in section 3.1. We hypothesize that the lesser change in γ with temperature for the LEV/XYL substrate compared to a GLY substrate (Gross et al. 2009), may be due to the viscosity difference between both substrates. Moise et al. (2002) studied the uptake of $NO_3$ by a saturated alcohol, 1-octanol, in the liquid and solid-phase. They derived $γ = 7.1\times10^{-3}$ at 258 K for the liquid phase of 1-octanol and $4.1\times10^{-3}$ at 248 K for the solid phase as shown in Table 2. The change in γ is a factor 1.73 over 10 K, mainly because of the rapid phase transition of 1-octanol. In our LEV/XYL uptake experiment, γ changes by a factor 2.74 over a temperature range of 80 K, however, γ measured here is about one order of magnitude lower. We suggest that the different sensitivity in γ over these different temperature ranges is

due to the continuous amorphous phase transition of the LEV/XYL mixture compared to the abrupt phase transition of 1-octanol.

At low temperatures, $\gamma$ for NO$_3$ uptake by LEV and LEV/XYL films are similar within our experimental uncertainties, as shown in Table 2. This is in contrast to expectations derived from the gas-phase structure activity relationships (SARs) (Atkinson, 1987;Kerdouci et al., 2014), which predict that the reaction NO$_3$ + LEV occurs 6.7 times faster than the reaction of NO$_3$ + XYL. Also, $\gamma$ of solid LEV and LEV/XYL is similar to NO$_3$ uptake by solid alkane substrates and monolayers, e.g. n-hexadecane given in Table 2 (Knopf et al., 2006;Knopf et al., 2011;Gross and Bertram, 2009;Moise et al., 2002). However, an alcohol is expected to be more reactive with NO$_3$ compared to an alkane, as the $\alpha$-hydrogen atom at the OH group is more labile and thus facilitates the hydrogen abstraction. These measurements indicate that the presence of labile $\alpha$-hydrogen atoms in solid substrate films do not significantly yield higher $\gamma$. A possible explanation may be that the number density of labile $\alpha$-hydrogen atoms at the interface is not sufficiently different between LEV, the LEV/XYL mixture and alkane substrates films. Furthermore, considering that NO$_3$ concentrations are too low to saturate the substrate surface, the probability of collisions with the most reactive hydrogen may also be low. This may result in similar $\gamma$ within our uncertainties, for both substrates. In conclusion, our study demonstrates a strong positive correlation between temperature, substrate phase state, and NO$_3$ uptake reactivity for saturated alcohols.

**3.5 Temperature modulated reactive OH uptake**

Figure 9a shows $\gamma$ of OH reacting with GLU as a function of temperature indicating no significant change in reactivity over the examined temperature range. As the temperature increases from 250 K on, $\gamma$ appears to slightly decrease from 0.11±0.03 to 0.09 ±0.03, although this change is within experimental uncertainties. GLU substrates maintain a solid or highly viscous phase state for all examined temperatures. The gas-phase structure activity relationship (SAR) predicts that the gas-phase reaction rate of OH radicals with GLU ($k_{Glu}^{g}$) decreases by a factor of 2.5 from 213 K to 298 K (Atkinson, 1987;Kwok and Atkinson, 1995). We also observe a decrease in reactivity (Fig. 9a), however, not as strong as during gas-phase reactions. This implies that other factors may also play a role in the observed heterogeneous reactivity. Since different chemical bonds have different reactivity and temperature dependence (Kwok and Atkinson, 1995), the reaction probability of OH radicals can be affected by the molecular orientation at the surface. SAR suggests that tertiary C-H bonds contribute to this negative temperature dependency of the reactivity. Hence, if more C-OH bonds are exposed to the surface compared to tertiary C-H bonds, this negative temperature dependency of reactivity weakens. A similar steric argument was given by Nah et al. (2014), who suggested that the increased surface density of C=C double bonds due to the formation of oleic acid dimers possibly leads to a faster surface reaction involving OH radicals. Also, the remaining amount of water in these substrates may impact the surface structure and the viscosity of the near-surface region, thereby potentially counteracting the decreasing reaction rate with increasing temperature predicted by SAR method.

For the GLU/HEX mixture (Fig. 9b), $\gamma$ at 303 K is 0.13±0.03, ~1.5 times larger than that at 233 K. The larger $\gamma$ at higher temperature can be attributed to the viscosity change of the substrate film with increasing temperature. $T_g$ is estimated as (248

± 3) K by our poke-flow experiment. Judging by the morphology and the flow characteristics of the organic film, it experiences a glass phase transition within the investigated temperature range. The major change in γ occurs above the expected $T_g$ of the GLU/HEX mixture. γ by GLU/HEX at low temperatures, i.e., between 213 K and 233 K, are lower than γ by GLU, and this is likely related to the lower reaction rate of HEX with OH radicals. For example, SAR predicts the reaction rate for HEX + OH to be 2.6 times slower than that for GLU + OH at 213 K.

Figure 9c shows that γ of CA increases by a factor of 5 within the experimental temperature range, from 0.13 ± 0.08 at 213 K to 0.66 ± 0.24 at 293 K. This increase of γ is particularly significant above the melting point of canola oil, $T_m = 258$ K (Fasina et al., 2008). Above the melting point, γ increases by a factor of ~2.4 between 273 K and 293 K, where the viscosity of canola oil changes from 0.185 to 0.079 Pa s (Fasina et al., 2006). Therefore, we attribute this large change in γ to the phase and viscosity change of the canola oil. We observed that as the canola oil is in a solid or viscous liquid (grease) state at 425 temperatures between 213 and 253 K, γ is less sensitive to temperature changes compared to when the canola oil is in a liquid phase. γ by CA is larger than γ by GLU at low temperatures although both are in solid phase, thus implying that OH reacts faster with C=C bonds than with C-H bonds and OH groups present on solid surfaces. Waring et al. (2011) measured the reactive loss of gas-phase OH radicals on squalene surfaces at room temperature, reporting γ = 0.39 ± 0.07. These measurements suggest that γ by CA can be larger than γ by squalene although squalene has more double bonds compared to 430 canola oil and both organics have similar viscosity (e.g., $\mu_{Sqe}$ is 0.012 Pa s at 298 K, Comunas et al. (2013)). Different accessibility of C=C bonds on the substrate surfaces could be one reason for different γ values. X-ray diffraction studies on liquid oleic acid showed that the unsaturated acid molecules exist primarily as dimers through hydrogen bonding of the carbonyl oxygen and the acidic hydrogen (Iwahashi and Kasahara, 2011;Iwahashi et al., 2000), and this possibly leads to more C=C double bonds at the substrate surface (Hearn et al., 2005). However, whether or not this phenomenon can compensate for 435 the more numerous C=C double bonds present in squalene molecules is uncertain. Lastly, gas-phase reaction kinetics suggest that unsaturated oxygenated organics (e.g., acids, alcohols, ketones) are more reactive toward OH radicals than their alkene equivalents due to the formation of a hydrogen bonded complex that facilities the formation of stable reaction products via interactions between the OH radical and the oxygenated functional group of the molecule (Mellouki et al., 2003;Orlando et al., 2001).

**4 Atmospheric implications**

Temperature varies greatly in the troposphere, both in the latitudinal and altitudinal directions. The zonal average surface temperature changes approximately 0.86 K per latitude degree in the northern hemisphere (Gates et al., 1999), and the vertical temperature gradient is typically defined by the lapse rate of ~10 K km$^{-1}$ for dry air and ~7 K km$^{-1}$ for wet air (Wallace and Hobbs, 2006). Globally, the temperature will greatly influence the phase state and the heterogeneous oxidation of OA particles, 445 especially for SOA particles that can exist a liquid phase state in the warmer planetary boundary layer but can be mostly solid in the colder middle and upper troposphere (Shiraiwa et al., 2017a).

The determined $\gamma$ values can be used to estimate the oxidation lifetime for a solid organic surface with the following equation (Moise and Rudich, 2001):

$$\tau = \frac{4N_{\text{tot}}}{\gamma \omega_X [X]_g} \quad , \tag{9}$$

where $\tau$ is the lifetime of one monolayer of coverage, i.e., how long it takes for 63% surface molecules to be oxidized. $N_{\text{tot}}$ represents the concentration of reaction sites on the organic substrate surface. Here, we apply this idealized approach to assess the degree of surface oxidation of OA particles in a solid or highly viscous phase state and when the oxidation reactions are confined to the surface. However, given that the surface-active organics are ubiquitous in tropospheric aerosols and organic films can exist on aerosol surfaces with potentially significant effects on atmospheric chemistry and climate, e.g.,

heterogeneous reactions, particle hygroscopicity, optical properties, and cloud forming activity (Jimenez et al., 2009;Kroll et al., 2015;Katrib et al., 2005b;Knopf et al., 2018;Slade et al., 2017;Robinson et al., 2007;Shiraiwa et al., 2017a;Cosman and Bertram, 2008;McNeill et al., 2006;Knopf et al., 2007;Knopf and Forrester, 2011;Moise et al., 2015) , this approach can further our understanding of the chemical evolution of atmospheric OA. We assume $N_{\text{tot}} = 10^{15}$ cm$^{-2}$ (Bertram et al., 2001). The concentration of gas oxidants O$_3$, NO$_3$, and OH are based on ambient conditions as $1\times10^{12}$, $1.25\times10^{9}$ and $1\times10^{6}$ molecules cm$^{-}$

$^3$ , respectively (Finlayson-Pitts and Pitts Jr, 1999). Figure 10 shows the effect of temperature on the surface species' lifetime for the different examined heterogeneous oxidation reactions. Oxidation of CA by O$_3$ proceeds within minutes to hours for typical tropospheric temperatures. Thus, degradation of unsaturated fatty acids is expected to proceed efficiently, even at colder temperatures. Despite the OH radical being the most effective oxidizer, Fig. 10 suggests that for middle and upper tropospheric conditions, oxidation of the particle surface by OH can take 1 to 2 weeks, emphasizing the slow physicochemical changes of

the particle properties during transport at high altitudes. However, closer to the surface, degradation can proceed almost an order of magnitude faster. Degradation of the particle surface by NO$_3$ may proceed by about a factor 2 slower at the coldest tropospheric temperatures compared to boundary layer conditions. Clearly, these datasets indicate that the topmost organic layers for most of the investigated OA surrogates can be oxidized within 1 week for lower and middle tropospheric conditions. However, as soon as OA particles reach higher altitudes and lower temperatures by, e.g., pyro-convection (Andreae et al.,

2004;Jost et al., 2004;Fromm and Servranckx, 2003), their atmospheric lifetime increases significantly. For example, aerosol particles originating from extreme wildfires like an Australia bushfire can circumnavigate the globe in weeks and can even reach the stratosphere existing for weeks or months (Ribeiro et al., 2020).

For OA particles, the lifetime ($\tau_p$) and particle degraded fraction (*DF*) can be estimated by the equations below (George et al., 2007):

$$\tau_p = \frac{4}{3} \frac{\rho R N_a}{\gamma \omega_X [X]_g M} \tag{10}$$

and

$$DF = 1 - \exp\left(-\frac{t}{\tau_p}\right), \tag{11}$$

where $R$ is the radius of the particle, chosen here as 100 nm, $N_a$ is Avagadro's number, $M$ is the molecular weight of the condensed-phase species, and $t$ is time. We derive $DF$ as a function of temperature and oxidant exposure time for oxidant

concentrations given above. However, we note that this $DF$ estimate is a simplified approach, since we assume that measured oxidation kinetics proceed throughout the entire particle with the same rate. This ignores the slow gas diffusion in the condensed phase and the hindered internal mixing of organic species, particularly when the OA particle is in a solid or highly viscous phase state. For OH oxidation, previous studies indicate that the oxidation reaction is confined to near the surface of a liquid or solid organic substrate, even for longer OH exposure periods at lower OH concentrations (Slade and Knopf,

2014;George and Abbatt, 2010;Lee and Wilson, 2016;Shiraiwa et al., 2011). However, whether bulk processes may significantly change the reactivity under long oxidant exposure as encountered in the atmosphere still needs to be examined. In the case of $O_3$ and $NO_3$ oxidation of organic substrates, the results by Shiraiwa et al. (2011) and Shiraiwa et al. (2012) indicate that the reactive uptake coefficients decrease with increasing exposure. Therefore, we probably underestimate the chemical lifetime. As such, the $DF$ values at lower temperatures likely represent upper limits. In other words, degradation in

ambient particles is expected to be less. Keeping this limitation in mind, Fig. 11 displays the estimated $DF$ for the examined oxidant-surrogate systems. As expected, for the lowest temperatures, the $DF$ values are lowest, implying longest lifetimes. The stronger the particle viscosity change with temperature, the greater is the change in $DF$, e.g., viscosity of CA decreases from $10^{12}$ to $10^{-2}$ Pa s over 213 to 293 K. This yields the largest change in $DF$ over this temperature range compared to the other investigated systems (considering the same exposure time period). Figures 11(c) and (d) display a small $DF$ within one month

for LEV/XYL and GLU/HEX mixtures in the upper troposphere. This is consistent with previous aircraft observations reporting a large amount of biomass burning aerosol particles in aged plumes at higher altitudes of the troposphere (Cubison et al., 2011). This also supports the hypothesis that aerosol particles originating from large wildfires (e.g., Australian bushfires) can be transported into the northern hemisphere from the stratosphere within a time period of one year (Deshler, 2008;Peterson et al., 2018). The presence of water vapor will significantly impact the phase state, in particular, of hygroscopic species such

as LEV and GLU (Zobrist et al., 2008;Koop et al., 2011;Mikhailov et al., 2009), where increasing humidity yields lower condensed-phase viscosity and, in turn, faster reaction kinetics (Slade and Knopf, 2014;Slade et al., 2017;Davies and Wilson, 2015). Neglecting this effect will lead to an underestimation of $DF$. This discussion neglects the chemical complexity of ambient OA where different condensed-phase species can result in different reactivities and reaction pathways (Zhang et al., 2015;Surratt et al., 2010;Ziemann and Atkinson, 2012;Knopf et al., 2005;Davies and Wilson, 2015). Those in turn can change

the multiphase kinetics and its dependency on temperature and particle phase state. Furthermore, heterogeneous particle composition and morphology can result in matrix effects or liquid-liquid phase separation, where, e.g., more reactive organic species are shielded by less reactive species (Lignell et al., 2014;Lee and Wilson, 2016;Charnawskas et al., 2017;Bertram et al., 2011). Those effects were not assessed in this study but necessitate additional experimental investigations.

## 5 Conclusions

In this study, we measured γ for several systems of oxidant and organic aerosol surrogate combinations including $O_3$ and OH + CA, $NO_3$ + LEV and LEV/XYL, OH + GLU and GLU/HEX, under dry conditions and for temperatures ranging from 213 K to 313 K. For the case of OH, this is the first low temperature reactive uptake study of which the authors are aware. The phase states of the organic substrate films were examined using the poke-flow technique allowing for an estimation of $T_g$ and substrate flow characteristics to constrain the magnitude of the substrate viscosity at different temperatures using the VTF
equation.

The strongest changes in heterogeneous reactivity observed for the examined oxidant-substrate systems correlates with the largest change in organic substrate viscosity with temperature associated with a solid-to-liquid phase transition. The largest reactivity occurs between $O_3$ and CA exhibiting a change by a factor of 34, due to the phase transition of CA. In general, we attribute the faster heterogeneous kinetics in the semi-solid and liquid phase state to surface and bulk reactions, the latter
enabled by increased diffusion coefficients of gas and condensed species resulting from lower viscosity. Furthermore, once in the liquid phase, as temperature increases viscosity decreases and diffusivity increases, leading to potentially strong increases in reactivity. LEV and GLU substrates display a semi-solid and solid phase state over the entire probed temperature range, with estimated viscosities larger than $10^4$ Pa s. As a result, the overall reactivity is lower compared to liquid substrate films and does not change significantly with temperature. Although viscosity in the semi-solid phase regime can change substantially
with temperature, viscosity can still be too high to allow for significant bulk processes to play a role. In this case, surface reactions likely dominate and replenishment of unoxidized molecules to the surface is hindered. Application of organic substrate mixtures to control $T_g$ to induce a solid-to-liquid phase transition as temperature increases, is accompanied with an increase in reactivity.

Our results are consistent with previous studies reporting the significance of particle phase state for the reactive uptake
kinetics (Arangio et al., 2015;Knopf et al., 2005;Kolesar et al., 2014;Slade and Knopf, 2014;Davies and Wilson, 2015;Shiraiwa et al., 2012). To resolve the molecular processes of the measured heterogeneous kinetics, detailed modelling studies (Pöschl et al., 2007;Shiraiwa et al., 2010;Arangio et al., 2015;Houle et al., 2015;Houle et al., 2018a;Pöschl and Shiraiwa, 2015) extended to lower temperatures are needed. A crucial aspect of this study is the interplay between the temperature dependence of the reaction kinetics and the desorption lifetime. At lower temperatures when an organic substrate in the solid state, over a
wide temperature range the reactivity does not change significantly. Desorption lifetime will likely increase significantly with decreasing temperature with subsequent effects on the reaction kinetics. Changes in substrate viscosity with temperature may also play a role in the overall heterogeneous kinetics when the substrate is in a semi-solid phase state. However, to what extent the particle viscosity will influence the diffusion and reaction kinetics is still not resolved. The comprehensive data set presented here will allow application of a more detailed kinetic multi-layer model to constrain the temperature dependency of
reaction and transport parameters. This study did not address the role of water vapor acting as a plasticizer concurrent to phase changes induced by temperature changes (Zobrist et al., 2008;Koop et al., 2011;Mikhailov et al., 2009). The role of humidity

on amorphous phase state and resulting multiphase kinetics has been studied at room temperature (Shiraiwa et al., 2011;Slade and Knopf, 2014;Davies and Wilson, 2015;Li et al., 2018). Most of these studies suggest that increasing humidity leads to faster reactive uptake kinetics. However, at lower temperatures, diffusivity is slower leading to kinetically hindered
adjustments of the condensed-phase state (Berkemeier et al., 2014;Knopf et al., 2018;Charnawskas et al., 2017;Wang et al., 2012). Future experimental studies should focus on how the coupled effects of ambient temperature and humidity on the amorphous phase state of OA particles modulate the multiphase oxidation kinetics.

Our study demonstrates unambiguously that the chemical reactivity of organic matter towards atmospheric oxidants can vary significantly in response to ambient temperature, which, in turn, modulates the organic phase state. Ambient OA, however,
display greater chemical and morphological complexity (Laskin et al., 2016;Laskin et al., 2019), and as such we expect varying multiphase reaction pathways having different reactivity towards atmospheric gas-phase oxidants which will translate into different reactivity dependencies on temperature and phase state. Despite this caveat, due to lower temperatures at higher altitudes, we can expect OA particles during transport in the free troposphere to have significantly longer lifetimes with respect to chemical degradation. As a result, we can expect OA particles during transport in the free troposphere to have significantly
longer lifetimes with respect to chemical degradation. This is important information for our understanding of the chemical evolution of OA particles and their impact on source apportionment, air quality, and climate.

## 6 Appendix: The impact of gas flow entrance effects and velocity profiles on the reactive uptake kinetics

The KPS method (Knopf et al., 2015), similar to the Cooney-Kim-Davis (CKD) method applied by Murphy and Fahey (1987) , accounts for gas flow entrance effects into the flow reactor, i.e., the establishment of concentration profiles, that impact the
derivation of the reactive uptake coefficient as pointed out by Davis (2008). When considering the establishment of gas concentration profiles for correction of observed pseudo-first order wall loss rates, slower uptake reaction kinetics (e.g., the uptake of $O_3$), typically represent conditions of either having a slow gas flow or long flow tube, and thus the $N_{shw}^{eff} \approx 3.66$ as shown in Fig. A1 (Davis, 2008). In this case, the different approaches (Brown, CKD, and KPS) to correct observed wall loss rate for transport limitations by diffusion yield the same γ values as illustrated in Fig. 7 (CKD approach is not shown but yields
same results as KPS). However, reactive OH uptake exerts faster reaction kinetics and thus resembles conditions of a fast gas flow or a short flow tube (Fig. A1) where $N_{shw}^{eff}$ can significantly depart from 3.66 (Knopf et al., 2015;Murphy and Fahey, 1987;Davis, 2008). In this case, γ values derived by the Brown method always yield larger values compared to those derived by the KPS method as illustrated in Fig. A2 for the uptake of OH by glucose. Hence, for fast uptake kinetics, we recommend using either the KPS or CKD methods to derive accurate uptake kinetics when using a coated-wall flow reactor.
The gas flow velocity ratio between mean flow velocities in the flow reactor ($u_{mean}^{ann}$) and the gas flow exiting the movable injector ($u_{mean}^{inj}$) also impacts the derivation of γ. If $u_{mean}^{ann} < u_{mean}^{inj}$, reactive uptake experiments are ambiguous, potentially resulting in an underestimation of the uptake kinetics, as pointed out by Davis (2008). This flow condition can lead to a jet-

like exit gas flow from the movable injector. Figure A2 displays this effect when $u_{\text{mean}}^{\text{ann}}/u_{\text{mean}}^{\text{inj}} < 1$, where the measured γ values can deviate by a factor of 3 from the actual value.

For all experimentally derived γ values, we set $u_{\text{mean}}^{\text{ann}}/u_{\text{mean}}^{\text{inj}} > 1.27$ for the flow tube with 1.758 cm inner diameter (ID) and $u_{\text{mean}}^{\text{ann}}/u_{\text{mean}}^{\text{inj}} > 1.33$ for the flow tube with an ID of 1.2 cm, based on the theoretical calculation below. The velocity profile and volume flow rate Q of a Poiseuille flow in the annular section can be described by the equations (Rosenhead, 1988)

$$u(r) = \frac{G}{4\mu}\left[(R_1^2 - r^2) + (R_2^2 - R_1^2)\frac{\ln(r/R_1)}{\ln(R_2/R_1)}\right] \tag{A1}$$

and

$$Q = \int_{R_1}^{R_2} 2\pi r u(r) dr = \frac{G\pi}{8\mu}\left[R_2^4 - R_1^4 - \frac{(R_2^2 - R_1^2)^2}{\ln(R_2/R_1)}\right] , \tag{A2}$$

where $u(r)$ is the flow velocity profile as a function of the radius $r$. $G$ is a constant pressure gradient, $\mu$ is the dynamic viscosity, $R_1$ is the inner cylinder radius, $R_2$ is the outer cylinder radius, and $r$ is a radius value between $R_1$ and $R_2$. If $R_1$, $R_2$ and constant parameters ($G$ and $\mu$) are known, the maximum flow velocity $u_{\text{max}}$ is obtained by setting $du(r)/dr = 0$, and the mean flow velocity in the annular section $u_{\text{mean}}$ is derived from $Q/A_{\text{ann}}$, where $A_{\text{ann}}$ is the cross-sectional area of the annular section.

Using the flow tube with a diameter of 1.758 cm as an example, and setting $R_1 = 0.325$ cm and $R_2 = 0.879$ cm, the ratio of maximum flow velocity over the average flow velocity in the annular section is $u_{\text{max}}^{\text{ann}}/u_{\text{mean}}^{\text{annul}} = 1.575$. The ratio of the mean flow velocity over the maximum velocity in a tubular injector is $u_{\text{max}}^{\text{inj}}/u_{\text{mean}}^{\text{inj}} = 2$ as the injector flow is laminar (Rogers, 1992). Therefore, even when adjusting the gas flows in the injector and flow reactor to yield the same mean flow velocities, the difference in the maximum flow velocities can still differ significantly. In this case, the difference can be up to 27%, potentially resulting in a jet-like gas flow profile. For this reason, we set $u_{\text{mean}}^{\text{ann}}/u_{\text{mean}}^{\text{inj}} > 1.27$ to avoid this effect. In the case of using a flow tube with a diameter of 1.2 cm, $u_{\text{mean}}^{\text{ann}}/u_{\text{mean}}^{\text{inj}} > 1.33$.

**Data availability**

Data and code generated from this study are available from the corresponding author upon request.

**Author contributions**

JL conducted O₃, NO₃, and OH uptake experiments. SF performed N₂O₅ synthesis and conducted NO₃ uptake experiments. JL conducted the poke-flow experiments. JL performed all analysis of data. SF conducted analysis of NO₃ uptake data and contributed to the writing of the manuscript. JL led the writing of the manuscript. DK oversaw the project, envisioned analysis, and contributed to the writing of the manuscript.

**Competing interest**

The authors declare no conflict of interest.

**Acknowledgements**

Support from the National Science Foundation grant AGS-1446286 is acknowledged. Partial support from the U.S. Department of Energy, Office of Science (BER), Atmospheric System Research (DE-SC0016370), is acknowledged.

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

**Table 1.** Estimated $T_g$ of the applied substrate films. $T_g^{\text{exp}}$ is the measured $T_g$ using the poke-flow technique. $T_g^{\text{lit}}$ is the literature reported $T_g$. $T_g^{\text{pred}}$ is predicted $T_g$ for mixtures by Gordon-Taylor equation using $T_g^{\text{lit}}$ and $k_{\text{GT}}$. $k_{\text{GT}}$ is solute specific constant in the Gordon-Taylor equation. D is fragility.

| Substrate | $T_g^{\text{exp}}$ / K | $T_g^{\text{pred}}$ / K | $T_g^{\text{lit}}$ / K | $k_{\text{GT}}$ | $D$ |
|---|---|---|---|---|---|
| Levoglucosan (LEV) | 243 ± 4 | | 248 ± 2 [a,h,g] | 3.26 [a] | 14.1 [b] |
| Xylitol (XYL) | | | 249 ± 7 [f,g,i] | 2.1 [c] | 8.65 [b] |
| LEV/XYL | 238 ± 3 | 249 ± 5 | | | 11.3 [d] |
| Glucose (GLU) | 273 ± 3 | | 305 ± 13 [e,g,j] | 3.95 [e] | 12.1 [b] |
| 1,2,6-Hexanetriol (HEX) | | | 204 ± 6 [f,g,i] | 0.88 [e] | 13.16 [f] |
| GLU/HEX | 248 ± 3 | 252 ± 7 | | | 12.3 [d] |

[a] Lienhard et al. (2012). [b] DeRieux et al. (2018). [c] Elamin et al. (2012). [d] Interpolated values based on mass fraction. [e] Zobrist et al. (2008). [f] Nakanishi et al. (2011). [g] Rothfuss (2019). [h] Tombari and Johari (2015). [i] Dorfmüller et al. (1979). [j] Diogo and Ramos (2008).





**Table 2.** Comparison of uptake coefficients of $NO_3$ on liquid and solid substrates of saturated and unsaturated organics.

| Surface | Liquid surface | | Solid surface | |
| --- | --- | --- | --- | --- |
| | $T$ / K | $\gamma_{liquid}$ | $T$ / K | $\gamma_{solid}$ |
| LEV | | | 293 | $(4.2\pm0.6)\times10^{-4}$[a] |
| LEV | | | 213 | $(2.8\pm0.3)\times10^{-4}$[a] |
| LEV/XYL | 293 | $(8\pm2)\times10^{-4}$[a] | 213 | $(3.10\pm0.9)\times10^{-4}$[a] |
| Glycerol | 293 | $(1.4\pm0.3)\times10^{-3}$[b] | | |
| Glycerol | 268 | $(8.3\pm0.5)\times10^{-4}$[b] | | |
| 1-octanol | 258 | $(7.1\pm1.6)\times10^{-3}$[c] | 248 | $(4.1\pm1.0)\times10^{-3}$[c] |
| DES | 298 | $(4.1\pm0.3)\times10^{-3}$[b] | 263 | $(2.5\pm0.2)\times10^{-4}$[b] |
| n-Hexadecane | 293 | $(2.6\pm0.8)\times10^{-3}$[c] | 283-289 | $(3.8\pm1.0)\times10^{-4}$[c] |

[a]This study. [b]Gross et al. (2009). [c]Moise et al. (2002).





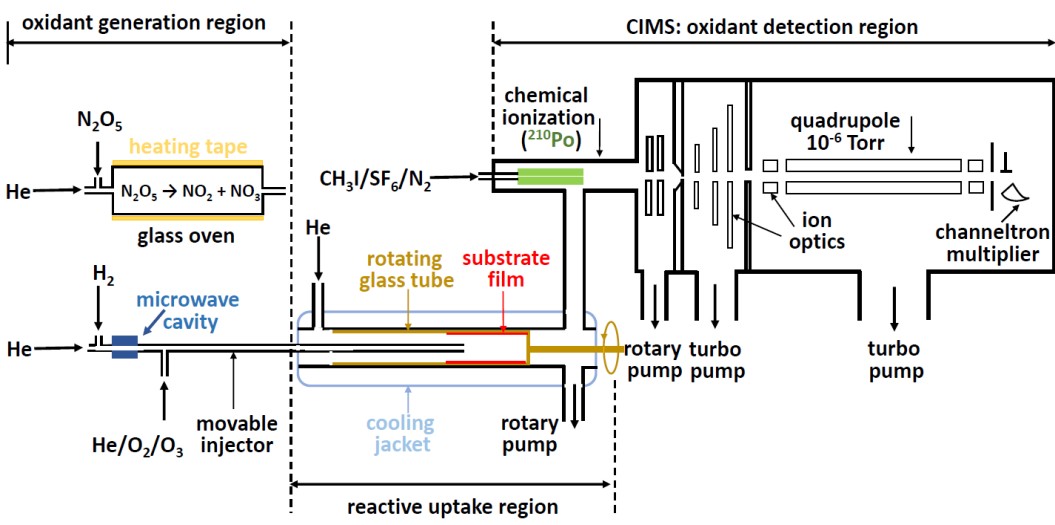

**Figure 1:** Schematic of the low-temperature coated-wall flow reactor coupled to the chemical ionization mass spectrometer (CIMS). The oxidant generation region displays the generation methods for $NO_3$ and OH radicals.





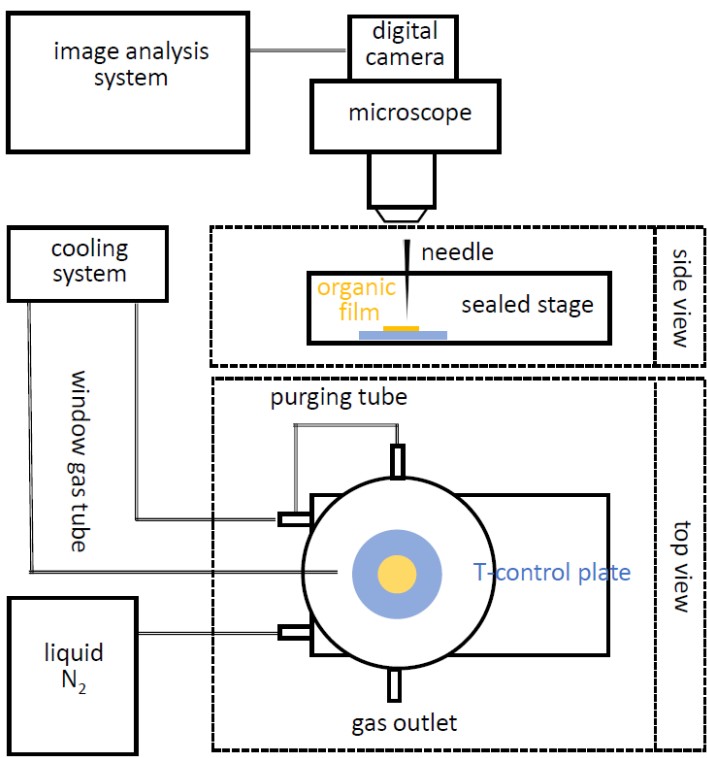

**Figure 2.** Schematic of the experimental poke-flow setup. This includes a temperature controlled Linkam cooling stage coupled to a microscope equipped with a digital camera and an image analysis system.




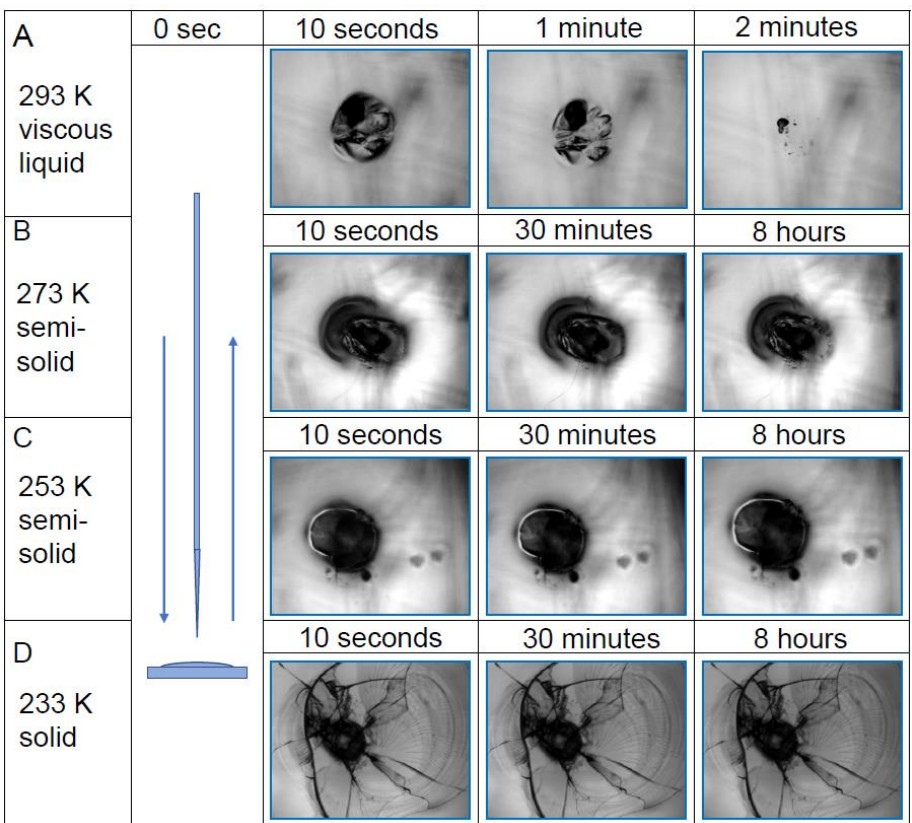

**Figure 3.** Estimation of the phase state of the film substrate as shown for a GLU/HEX mixture (mass ratio of 4:1). Deformation and recovery were monitored for different time periods due to the poking of the substrate at different temperatures. The microscope images are 200 μm wide.



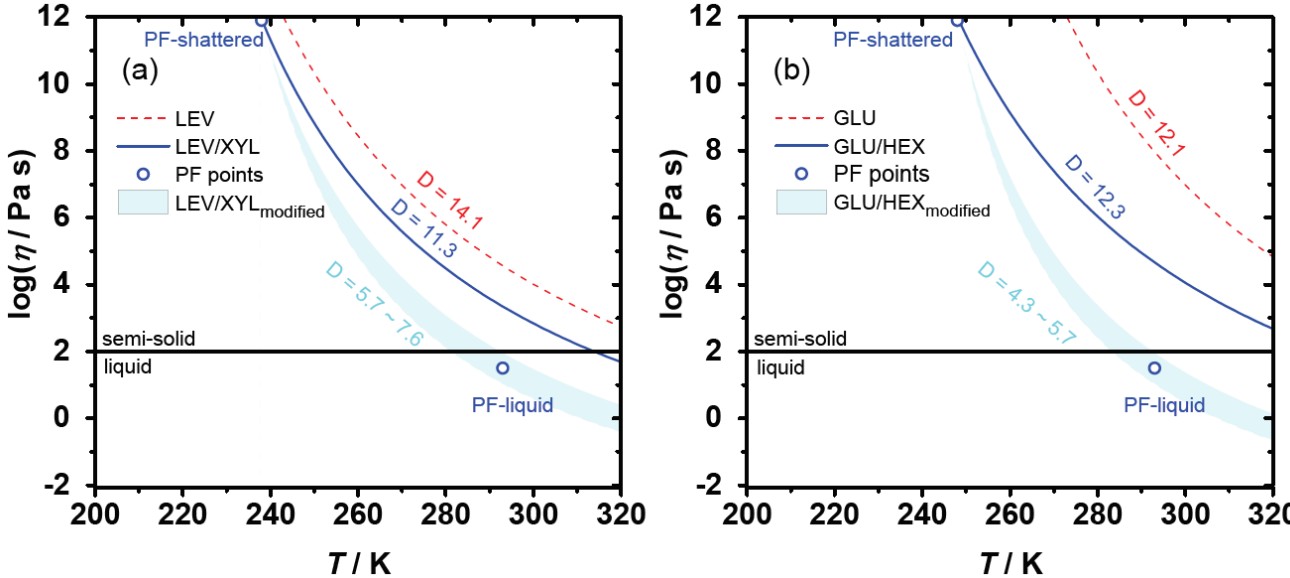

**Figure 4.** The Angell plot of viscosity as a function of temperature for LEV, LEV/XYL, GLU and GLU/HEX substrates. The lines represent predicted viscosities as a function of $T_g^{exp}$ and applied $D$ values (see Table 1). PF-shattered and PF-liquid (blue circles) indicate the conditions for which the poke-flow (PF) experiment detected a solid (glassy) and liquid phase state of the substrate, respectively. The shaded areas represent predicted viscosities by using the VTF equation with modified $D$ values as indicated in the panels based on the estimated viscosities derived from the poke-flow experiment. The horizontal black line at $\eta = 10^2$ Pa s indicates the threshold of liquid and semi-solid phase states.




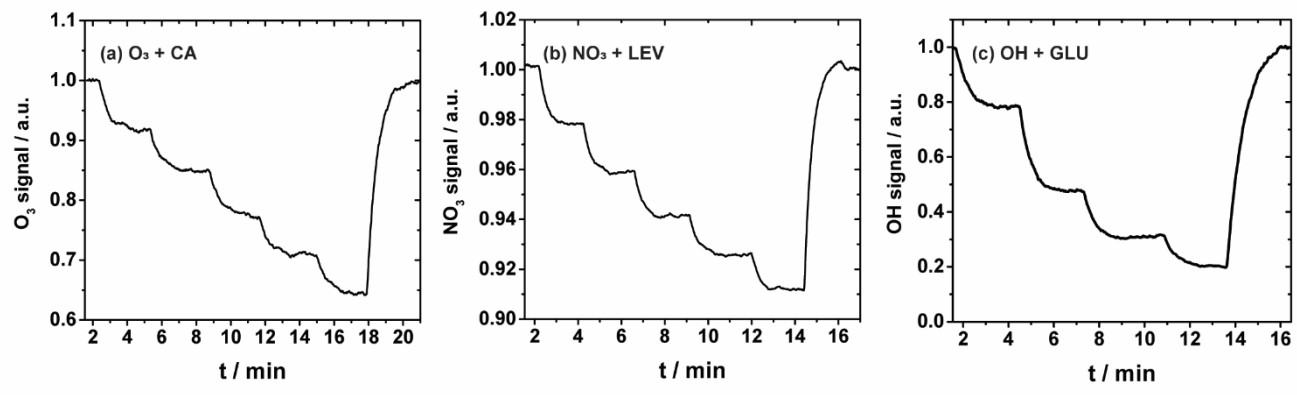

**Figure 5.** Reactive uptake experiments showing the change in the normalized gas-phase oxidant signal as the reaction time is changed by pulling back the injector incrementally. (a) uptake of $O_3$ by canola oil (CA); (b) uptake of $NO_3$ by levoglucosan (LEV); (c) uptake of OH radical by glucose (GLU).





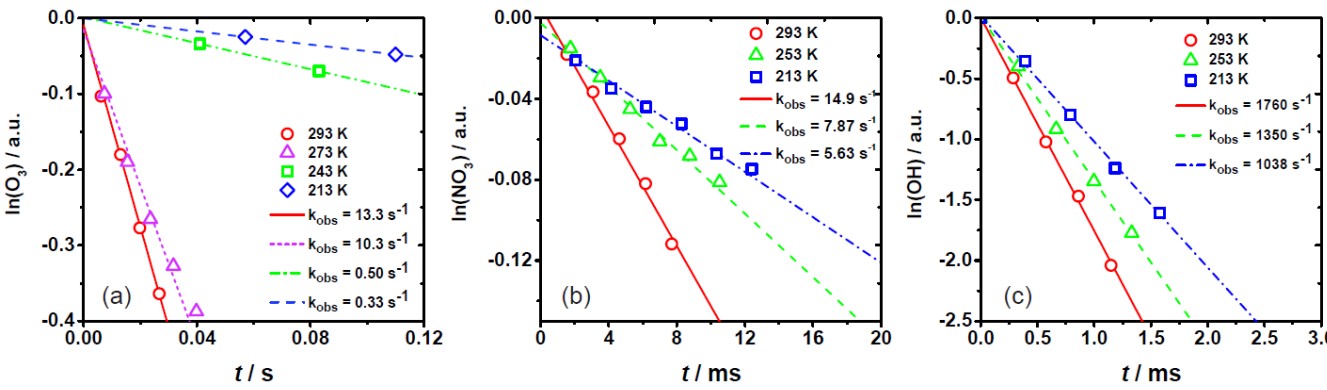

**Figure 6.** Natural logarithm of the change in gas-phase oxidant signal as a function of reaction time for (a) $O_3$ uptake by CA, (b) $NO_3$ uptake by LEV/XYL mixture, and (c) OH uptake by GLU/HEX mixture. Open circles, triangles, squares, and diamonds correspond to uptake measurements at different temperatures. Lines represent the corresponding linear fits to the data and corresponding slopes. $k_{obs}$ values are given in legend.

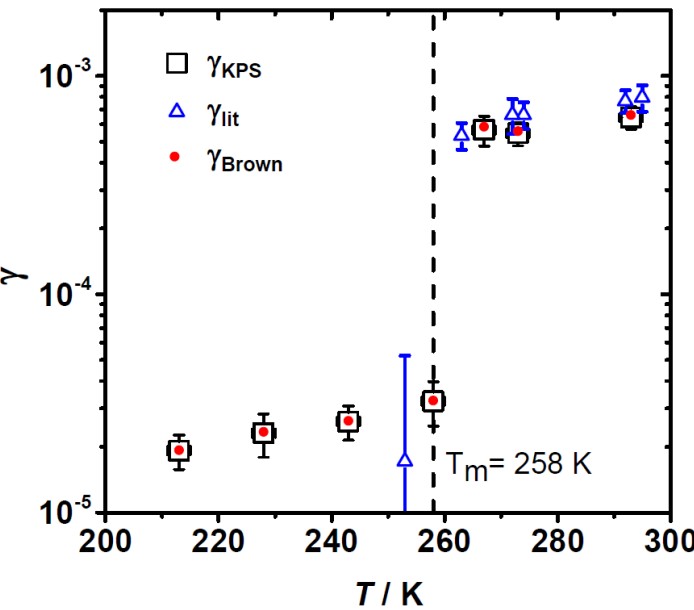

**Figure 7.** Reactive uptake coefficients of O$_3$ by canola oil as a function of temperature. Squares and red circles represent γ values derived using the KPS and Brown method, respectively. See text for more details. **γ$_{lit}$** represents values reported by de Gouw et al. (1998).

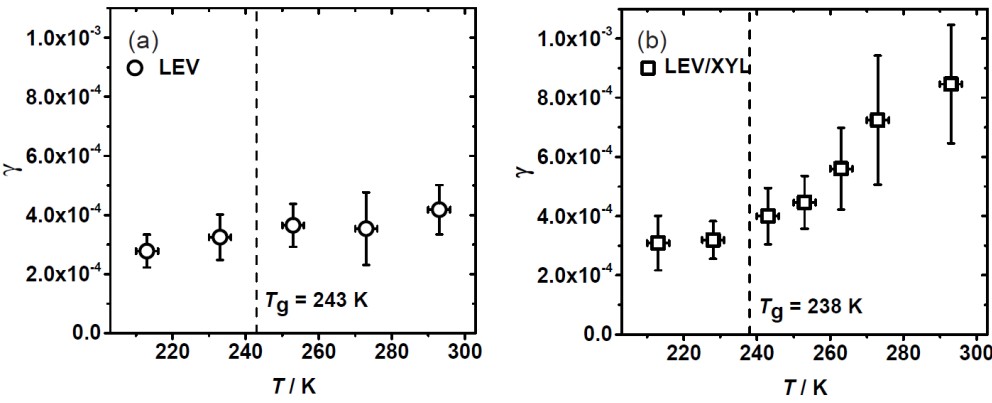

**Figure 8.** Reactive uptake coefficients, γ, of $NO_3$ reacting with (a) levoglucosan (LEV) and (b) 1:1 by mass LEV and xylitol (XYL) mixture as a function of temperature. The dashed line represents the glass transition temperature ($T_g$) measured in the poke-flow experiment.





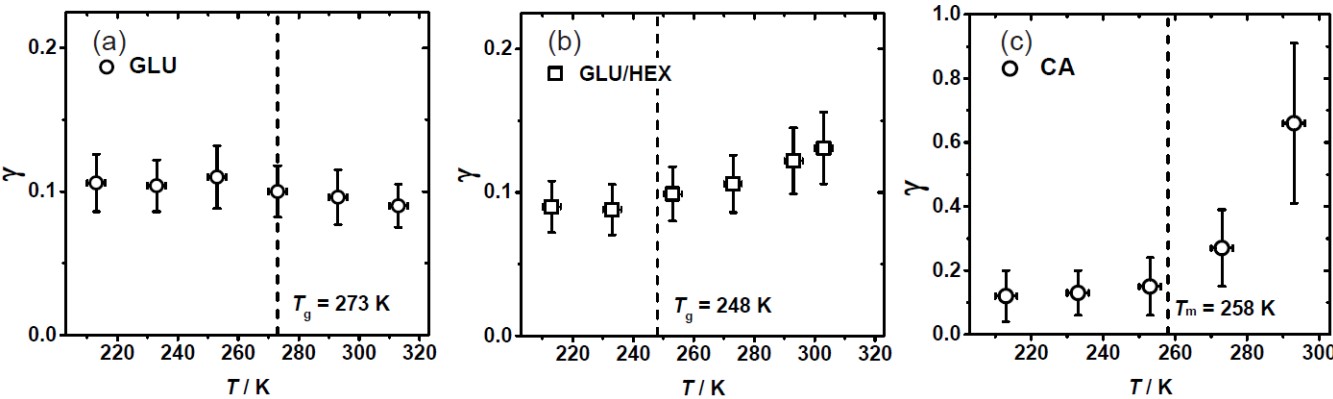

**Figure 9.** Reactive uptake coefficients, $\gamma$, of OH radicals reacting with (a) glucose (GLU), (b) 1:4 by mass glucose and 1,2,6-hexanetriol (GLU/HEX) mixture and (c) canola oil (CA) as a function of temperature. $T_g$ is the glass transition temperature determined in the poke-flow experiment.





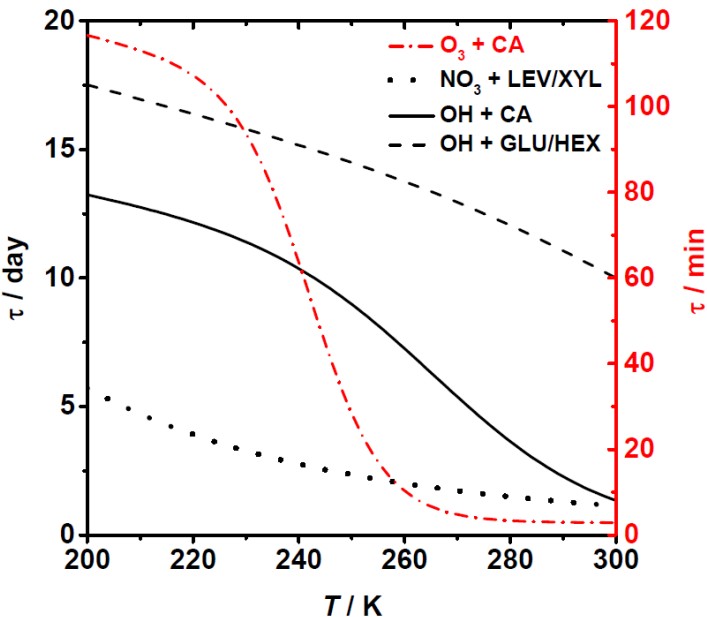

**Figure 10.** Lifetime estimates of one monolayer of examined OA surrogates for typical background concentrations of $O_3$, $NO_3$, and OH. See text for more details.

1290

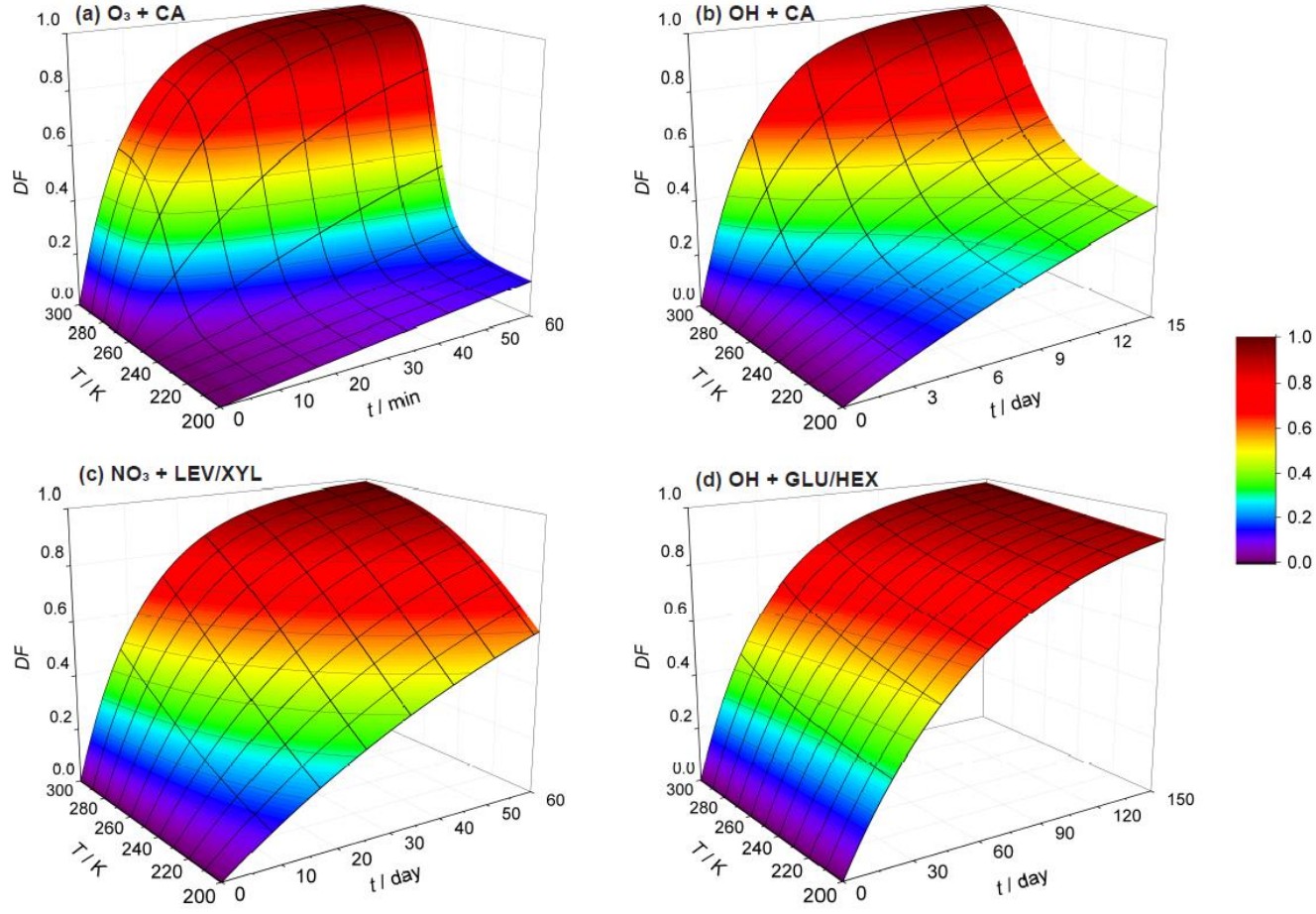

**Figure 11.** Exposure and temperature dependent particle degraded fraction (DF) for particles 200 nm in diameter for (a) $O_3$ oxidation of canola oil, (b) OH oxidation of canola oil, (c) $NO_3$ oxidation of LEV/XYL mixture, and (d) OH oxidation of GLU/HEX mixture. Note the different scales on the time axis.

1295

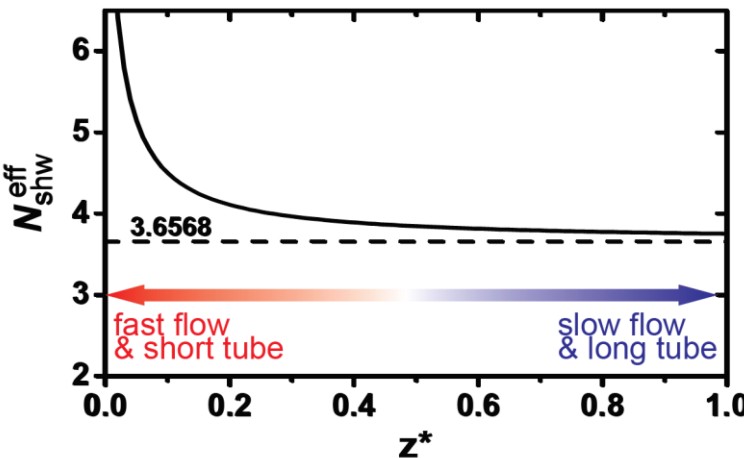

**Figure A1.** Dependence of the effective Sherwood number ($N_{shw}^{eff}$) on the dimensionless axial distance of the flow reactor. Smaller dimensionless axial distance represents the scenario of a fast flow or short tube. Adapted from Davies (2008) and Knopf et al. (2015).

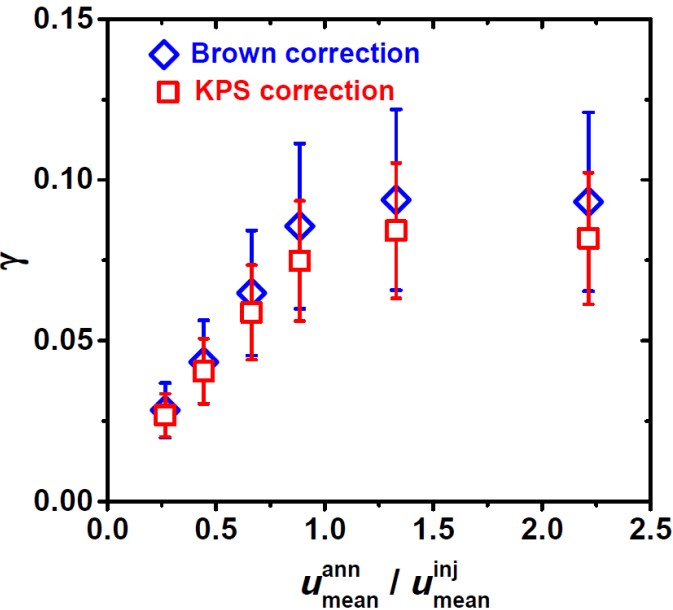

**Figure A2.** Differences in reactive uptake coefficient ($\gamma$) derived from two gas transport correction methods for fast heterogeneous kinetics involving OH uptake by glucose as a function of flow velocity ratio between mean gas flow velocity in the annular section of the flow reactor ($u_{mean}^{ann}$) and the gas flow exiting the movable injector ($u_{mean}^{inj}$). Red squares and blue diamonds represent KPS and Brown methods, respectively.

1325