# Peer review of "Heterogeneous oxidation of amorphous organic aerosol surrogates by O3, NO3, and OH at typical tropospheric temperatures"

_Atmospheric Chemistry and Physics, 2020_

## Referee Comment (RC1) · Anonymous Referee #1 · 9 Mar 2020

In this work, Li et al. investigated the reactive uptake coefficients of O3, NO3 and OH onto thin film of surrogate organic aerosols (OA) species over a troposphere-relevant temperature range. The underlying hypothesis is that the temperature induced change in phase state of the studied chemicals will impact the uptake kinetics of atmospheric oxidants onto OA and consequentially the degradation rate of OA. This was clearly shown from the measurement of uptake coefficient that took place in a temperature-controlled flow reactor with inner wall coated with the studied OA surrogate species. The phase state changes were supported by poke-flow experiments. The measured uptake coefficients onto the thin film were extended to estimate the temperature dependent particle degradation fraction (DF) for studied species/binary mixture as 200 nm

particles. Atmospheric implications were drawn that the lifetimes of OA are expected to be longer against heterogeneous oxidation due to a liquid-to-solid or semi-solid to solid phase transition that slows down the reactions. The overall experimentations were well thought and designed to minimize effects that can impact the accurate determination of reaction kinetics. The results were presented in high quality and discussed in detail with appropriate reference to the literature for data interpretation and comparison. Some specific comments provided below:

1. Could authors give more background to justify the choice of canola oil, levoglucosan and glucose as OA surrogates and the choice of paired oxidant as well?

2. There are no clear indications on the morphologies of films before being poked. Please consider adding these images to help interpret the images including those in the SI. Are the films for all individual species and mixtures supposed to be transparent before poking?

3. Line 224, would the Tg pred agree better with Tg exp if 10%-16% residual water was accounted for in the calculation? This will help to support the argument here.

4. In Section "Atmospheric implications" and "Conclusions", author should probably note that when coupled with higher ambient relative humidity the estimated temperature dependent DF using uptake coefficient derived from dry experiments may be underestimated for hygroscopic species like levoglucosan and glucose. Some discussions in this regard and directions for future studies could be added.

Technical Notes: 1. Font types of header and numbers in Table S4-Table S6 are not consistent.

---

## Referee Comment (RC2) · Anonymous Referee #2 · 22 Mar 2020

**Summary**
The manuscript by Li et al. used both experimental techniques and different mode simulations (including a viscosity model and a reaction kinetics model) to systematically understand the influence of temperature on the multiphase reactive process of $O_3$, $NO_3$, and OH on levoglucosan, xylitol, glucose, 1,2,3-hexanetriol, and canola oil mixtures. The authors carefully designed a flow tube reactor to measure the reactive uptake coefficient, $\gamma$, of these three gas phase species with organic thin films from 213 to 293 K. The results suggest that the phase state of the organic compounds at cooler temperatures in the upper troposphere could heavily impact the multiphase reactions and the lifetime of the organic aerosols.

The experimental section and the kinetic model sections of the manuscript are well done. The results are very important to help understand the multiphase process under low temperature regimes. The interpretation of the experimental data is a little bit unclear and overstated and should be corrected in the revision. With the following points being addressed, this manuscript is suitable to be published in Atmospheric Chemistry and Physics.

**Major Comment**

1. The author measured the reactive uptake coefficients of several species for residence times between 2 ms to 100 ms. Is the residence time safe to extrapolate the experimentally derived reactive uptake coefficients to ambient conditions where the residence time of the particles can be much longer? Shiraiwa et al. (2011) shows that the reactive uptake coefficients of ozone can vary significantly depending on the residence time. Can the author discuss about the potentially effects of short residence times in the manuscript?

2. The trends of the $\gamma$ values as a function of temperature are really interesting and it shows strong heterogeneity among different species. The authors seem to imply in the manuscript that phase state change is the dominant reason for $\gamma$ values to change (line 18, line 484-485, line 496-497). However, Shiraiwa et al. (2011&2012) demonstrate that if the organic species change their phase state from liquid to glass, the mass accommodation coefficients will change by orders of magnitude rather than just 34 times. Could the authors comment on the discrepancy between previous modeling results showing the uptake could change by orders of magnitude vs. the current experimental data showing the $\gamma$ values only change up to a factor of 34?

With carefully designed experiments, I do believe the $\gamma$ values the authors presented are accurate, but I think the data interpretation part is a little bit overstated to conclude phase state is the main reason for $\gamma$ to change. The authors need to discuss more why $\gamma$ values did not decrease as fast as previous modeling results (Shiraiwa et al., 2011&2012) shown when the viscosity changes. For instance, could the desorption lifetime (line 331) help increase $\gamma$ values as temperature decreases? I think it is important to discuss these aspects in the paper to show that maybe both phase state and the desorption kinetics play important, but counter roles, with the role of the phase state dominating the $\gamma$ value during this process.

3. The authors performed the poke-flow technique that was used in other papers to determine the glass transition temperatures of organic species. However, past literature on poke-flow technique were mostly used to determine the viscosity of the organic species and glass shattering phenomenon can indicate the viscosity is anywhere $>10^{12}$ Pa s (Lindsay et al. 2013), but does not necessarily mean the viscosity is exactly near $10^{12}$ Pa s when the compound is right around glass transition. The accuracy of the glass transition temperature derived from this technique will heavily impact data interpretation in Fig. 7-9. Given that the $T_g$ values shown in Table 1 do not match the $T_g$ values from literature (and the author also mentioned these compounds may have water in them, so it is difficult to compare the $T_g$ values with pure compounds from the literature), can the author provide any evidence to consistently show that the poke-flow technique can be used to derived accurate $T_g$ values? Given there hasn't been studies before using observation on glass shattering to solely determine the glass transition temperature, maybe the author can cross compare with previous data to validate this technique (such as Koop et al. 2010, Dette et al. 2015).

4. What is the relative humidity of the flow tube when the experiments were performed? Given the author performed the flow experiments under a variety of low temperatures, even small partial pressure of water could potential lead to relatively high humidity and alter the phase state of the organic film and increase the error bar range of the estimated viscosity. Another related question is that the poke flow measurement also was performed under "sealed room air" (line 190). The water vapor in the room air may lead to near high relative humidity at cooler temperatures. How would the authors ensure the RH levels in the flow tube and the poke flow cooling state are the same and don't lead to errors in estimating the phase state?

5. The author used "slight changes in viscosity" (line 330) and "small changes in viscosity" (line 491) multiple times in the manuscript to describe viscosity change as temperature drops. However, the changes in viscosity during semi-solid phase state spans over several orders of magnitude (Figure 4) and I would not consider these changes small. In fact, these changes coule be even larger than the phase state transition from liquid to semi-solid. This leads to my question when viscosity changes a few orders of magnitude from liquid to semi-solid, it seems the uptake coefficients are still the same.

6. The logical deductions of how phase state alters $\gamma$ values are a bit confusing and somewhat contradictory in multiple parts of the manuscript. For instance, line 339 shows the largest change of $\gamma$ is observed around the expected $T_g$ for LEV/XYL film, showing a transition from semi-solid to glass regime has the maximum influence of $\gamma$. However, in the next sentence, the author made a contradictory comment (line 340-341), saying "significant change in $NO_3$ uptake reactivity in the investigated temperature range can be attributed to the transition of a liquid to a highly viscous or solid substrate film". The similar statement appeared in line 351-352 saying that "the lower sensitivity of $\gamma$ on temperature may be due to the generally higher viscosity" of the compound studied. Again, in line 358-400 the authors state that the $\gamma$ is less sensitive to temperature change when canola is in semi-solid/glass phase states. If the $\gamma$ change is less obvious, then why would the author observe the largest change of $\gamma$ around $T_g$ for LEV/XYL film?

The authors also compared the γ change with the same amount of temperature change (25 K) at higher temperature regimes (liquid) and lower temperature regimes (semi-solid/solid) in the manuscript. Results show that that γ changes more significantly at higher temperature regimes (liquid). My question is, would the statement still be correct when comparing viscosity change instead of temperature change? I.e., will γ still change more significantly when viscosity changes the same orders of magnitude from liquid to semi-solid vs. semi-solid to glass?

7. The reactive uptake data as a function of temperature from this manuscript is very insightful since there have not been such data available at low temperature. Results also show that change of γ as a function of temperature is drastically different when it comes to different organic species. For instance, OH oxidation of GLU/HEX mix and $NO_3$ oxidation of LEV is not sensitive to temperature while OH and $O_3$ oxidation oxidations of canola oil are more sensitive to temperature change. Can the author discuss further in the manuscript why different species exhibit difference in γ value changes? Given the strong heterogeneity of how the γ values of different compounds respond to temperature change, can the author still make the conclusions that "chemical reactivity of organic matter towards atmospheric oxidants can vary significantly in response to ambient temperature", as the ambient OA may have a different response of γ towards temperature than the organic surrogate compounds used in this study?

**Minor Comments**

1. Line 287: The equation doesn't seem to be right. Why would the diffusion coefficient to the left side of the equation be in the numerator, but the diffusion coefficients to the right side of the equation be in the denominator? Could the authors please double check?

2. The literature citation in the introduction part is good overall, but missed some recent important publications that are highly relevant to this study. Please include those citations for a more comprehensive introduction.

Line 40: please add the following two papers that are relevant.
Murray, B. J., T. W. Wilson, S. Dobbie, Z. Cui, S. M. R. K. Al-Jumur, O. Mohler, M. Schnaiter, R. Wagner, S. Benz, M. Niemand, H. Saathoff, V. Ebert, S. Wagner and B. Karcher (2010). "Heterogeneous nucleation of ice particles on glassy aerosols under cirrus conditions." Nature Geosci **3**(4): 233-237.

Pöschl, U. and M. Shiraiwa (2015). "Multiphase Chemistry at the Atmosphere–Biosphere Interface Influencing Climate and Public Health in the Anthropocene." Chemical Reviews **115**(10): 4440-4475.

Line 42: please add the following two papers that are relevant.
Renbaum-Wolff, L., J. W. Grayson, A. P. Bateman, M. Kuwata, M. Sellier, B. J. Murray, J. E. Shilling, S. T. Martin and A. K. Bertram (2013). "Viscosity of α-pinene secondary organic material and implications for particle growth and reactivity." Proceedings of the National Academy of Sciences **110**(20): 8014-8019.

Kidd, C., V. Perraud, L. M. Wingen and B. J. Finlayson-Pitts (2014). "Integrating phase and composition of secondary organic aerosol from the ozonolysis of α-pinene." Proceedings of the National Academy of Sciences **111**(21): 7552-7557.

Line 44-45: please include the following two papers that are relevant.
Dette, H. P. and T. Koop (2015). "Glass Formation Processes in Mixed Inorganic/Organic Aerosol Particles." The Journal of Physical Chemistry A **119**(19): 4552-4561.

Zhang, Y., L. Nichman, P. Spencer, J. I. Jung, A. Lee, B. K. Heffernan, A. Gold, Z. Zhang, Y. Chen, M. R. Canagaratna, J. T. Jayne, D. R. Worsnop, T. B. Onasch, J. D. Surratt, D. Chandler, P. Davidovits and C. E. Kolb (2019). "The Cooling Rate- and Volatility-Dependent Glass-Forming Properties of Organic Aerosols Measured by Broadband Dielectric Spectroscopy." Environmental Science & Technology **53**(21): 12366-12378.

Line 47: please include the following three papers that are relevant
Rothfuss, N. E. and M. D. Petters (2016). "Coalescence-based assessment of aerosol phase state using dimers prepared through a dual-differential mobility analyzer technique." Aerosol Science and Technology **50**(12): 1294-1305.

Zhang, Y., M. S. Sanchez, C. Douet, Y. Wang, A. P. Bateman, Z. Gong, M. Kuwata, L. Renbaum-Wolff, B. B. Sato, P. F. Liu, A. K. Bertram, F. M. Geiger and S. T. Martin (2015). "Changing shapes and implied viscosities of suspended submicron particles." Atmos. Chem. Phys. **15**(14): 7819-7829.

Järvinen, E., K. Ignatius, L. Nichman, T. B. Kristensen, C. Fuchs, C. R. Hoyle, N. Höppel, J. C. Corbin, J. Craven, J. Duplissy, S. Ehrhart, I. El Haddad, C. Frege, H. Gordon, T. Jokinen, P. Kallinger, J. Kirkby, A. Kiselev, K. H. Naumann, T. Petäjä, T. Pinterich, A. S. H. Prevot, H. Saathoff, T. Schiebel, K. Sengupta, M. Simon, J. G. Slowik, J. Tröstl, A. Virtanen, P. Vochezer, S. Vogt, A. C. Wagner, R. Wagner, C. Williamson, P. M. Winkler, C. Yan, U. Baltensperger, N. M. Donahue, R. C. Flagan, M. Gallagher, A. Hansel, M. Kulmala, F. Stratmann, D. R. Worsnop, O. Möhler, T. Leisner and M. Schnaiter (2016). "Observation of viscosity transition in α-pinene secondary organic aerosol." Atmos. Chem. Phys. **16**(7): 4423-4438.

Line 51: please add the following two relevant papers:
Shiraiwa, M. and J. H. Seinfeld (2012). "Equilibration timescale of atmospheric secondary organic aerosol partitioning." Geophysical Research Letters **39**(24): L24801.

Renbaum-Wolff, L., J. W. Grayson, A. P. Bateman, M. Kuwata, M. Sellier, B. J. Murray, J. E. Shilling, S. T. Martin and A. K. Bertram (2013). "Viscosity of α-pinene secondary organic material and implications for particle growth and reactivity." Proceedings of the National Academy of Sciences **110**(20): 8014-8019.

Line 53: please include the following two papers that are relevant:
Gaston, C. J., J. A. Thornton and N. L. Ng (2014). "Reactive uptake of N2O5 to internally mixed inorganic and organic particles: the role of organic carbon oxidation state and inferred organic phase separations." Atmos. Chem. Phys. **14**(11): 5693-5707.

Zhang, Y., Y. Chen, A. T. Lambe, N. E. Olson, Z. Lei, R. L. Craig, Z. Zhang, A. Gold, T. B. Onasch, J. T. Jayne, D. R. Worsnop, C. J. Gaston, J. A. Thornton, W. Vizuete, A. P. Ault and J. D. Surratt (2018). "Effect of Aerosol-Phase State on Secondary Organic Aerosol Formation from the Reactive Uptake of Isoprene-Derived Epoxydiols (IEPOX)." Environmental Science & Technology Letters **5**(3): 167-174.

Line 55: Please add the following paper that is relevant:
Riedel, T. P., Y.-H. Lin, S. H. Budisulistiorini, C. J. Gaston, J. A. Thornton, Z. Zhang, W. Vizuete, A. Gold and J. D. Surratt (2015). "Heterogeneous Reactions of Isoprene-Derived Epoxides: Reaction Probabilities and Molar Secondary Organic Aerosol Yield Estimates." Environmental Science & Technology Letters **2**(2): 38-42.

Line 61: please include the following citations that are highly relevant to the paper:
Pajunoja, A., W. Hu, Y. J. Leong, N. F. Taylor, P. Miettinen, B. B. Palm, S. Mikkonen, D. R. Collins, J. L. Jimenez and A. Virtanen (2016). "Phase state of ambient aerosol linked with water uptake and chemical aging in the southeastern US." Atmos. Chem. Phys. **16**(17): 11163-11176.

Gaston, C. J., J. A. Thornton and N. L. Ng (2014). "Reactive uptake of N2O5 to internally mixed inorganic and organic particles: the role of organic carbon oxidation state and inferred organic phase separations." Atmos. Chem. Phys. **14**(11): 5693-5707.

Houle, F. A., A. A. Wiegel and K. R. Wilson (2018). "Predicting Aerosol Reactivity Across Scales: from the Laboratory to the Atmosphere." Environmental Science & Technology **52**(23): 13774-13781.

Zhang, Y., Y. Chen, A. T. Lambe, N. E. Olson, Z. Lei, R. L. Craig, Z. Zhang, A. Gold, T. B. Onasch, J. T. Jayne, D. R. Worsnop, C. J. Gaston, J. A. Thornton, W. Vizuete, A. P. Ault and J. D. Surratt (2018). "Effect of Aerosol-Phase State on Secondary Organic Aerosol Formation from the Reactive Uptake of Isoprene-Derived Epoxydiols (IEPOX)." Environmental Science & Technology Letters **5**(3): 167-174.

Riva, M., et al. (2019). "Increasing Isoprene Epoxydiol-to-Inorganic Sulfate Aerosol (IEPOX:Sulfinorg) Ratio Results in Extensive Conversion of Inorganic Sulfate to Organosulfur Forms: Implications for Aerosol Physicochemical Properties." Environmental Science & Technology **53**(15): 8682-8694.

Zhang, Y., Y. Chen, Z. Lei, N. E. Olson, M. Riva, A. R. Koss, Z. Zhang, A. Gold, J. T. Jayne, D. R. Worsnop, T. B. Onasch, J. H. Kroll, B. J. Turpin, A. P. Ault and J. D. Surratt (2019). "Joint Impacts of Acidity and Viscosity on the Formation of Secondary Organic Aerosol from Isoprene Epoxydiols (IEPOX) in Phase Separated Particles." ACS Earth and Space Chemistry **3**(12): 2646-2658.

DeRieux, W.-S. W., P. S. J. Lakey, Y. Chu, C. K. Chan, H. S. Glicker, J. N. Smith, A. Zuend and M. Shiraiwa (2019). "Effects of Phase State and Phase Separation on Dimethylamine Uptake of

Ammonium Sulfate and Ammonium Sulfate–Sucrose Mixed Particles." ACS Earth and Space Chemistry **3**(7): 1268-1278.

Line 64: please include the following citation that is relevant:
Arangio, A. M., J. H. Slade, T. Berkemeier, U. Pöschl, D. A. Knopf and M. Shiraiwa (2015). "Multiphase chemical kinetics of OH radical uptake by molecular organic markers of biomass burning aerosols: humidity and temperature dependence, surface reaction, and bulk diffusion." The Journal of Physical Chemistry A **119**(19): 4533-4544.

Line 70-73: there are other works that also analyzed the effects of phase states on OH oxidation that should be included:
Houle, F. A., A. A. Wiegel and K. R. Wilson (2018). "Predicting Aerosol Reactivity Across Scales: from the Laboratory to the Atmosphere." Environmental Science & Technology **52**(23): 13774-13781.
Houle, F. A., A. A. Wiegel and K. R. Wilson (2018). "Changes in Reactivity as Chemistry Becomes Confined to an Interface. The Case of Free Radical Oxidation of C30H62 Alkane by OH." The Journal of Physical Chemistry Letters **9**(5): 1053-1057.

3. The measured $T_g$ values of a few species listed in Table 1 are lower than the value measured by other literature. The author explains that was due to water remaining in the organic compound which lowers the glass transition temperature. If that is the case, it is a bit misleading to still associate the experimentally derived glass transition temperatures with the name of the pure compounds. Maybe the author should add information to highlight that the experimental derived $T_g$ values in Table 1 refers to the mixture of water and organic species.

4. n-Hexadecane in Table 2 was never mentioned in the main text. Please call out the name. Also, in Table 2, a comma should be added between reference number and the γ values

**Additional References:**
Shiraiwa, M., M. Ammann, T. Koop and U. Pöschl (2011). "Gas uptake and chemical aging of semisolid organic aerosol particles." Proceedings of the National Academy of Sciences **108**(27): 11003-11008.

Koop, T., J. Bookhold, M. Shiraiwa and U. Poschl (2011). "Glass transition and phase state of organic compounds: dependency on molecular properties and implications for secondary organic aerosols in the atmosphere." Physical Chemistry Chemical Physics **13**(43): 19238-19255.

Shiraiwa, M. and J. H. Seinfeld (2012). "Equilibration timescale of atmospheric secondary organic aerosol partitioning." Geophysical Research Letters **39**(24): L24801.

Renbaum-Wolff, L., J. W. Grayson, A. P. Bateman, M. Kuwata, M. Sellier, B. J. Murray, J. E. Shilling, S. T. Martin and A. K. Bertram (2013). "Viscosity of α-pinene secondary organic material and implications for particle growth and reactivity." Proceedings of the National Academy of Sciences **110**(20): 8014-8019.

Dette, H. P., M. Qi, D. C. Schröder, A. Godt and T. Koop (2014). "Glass-Forming Properties of 3-Methylbutane-1,2,3-tricarboxylic Acid and Its Mixtures with Water and Pinonic Acid." The Journal of Physical Chemistry A **118**(34): 7024-7033.

---

## Author Comment (AC1) · 26 Mar 2020

We thank the reviewer for evaluating our manuscript and giving suggestions for improvement. Following the reviewer' suggestions, we have revised our manuscript correspondingly. Listed below are our point-by-point responses given in normal font.

*In this work, Li et al. investigated the reactive uptake coefficients of O3, NO3 and OH onto thin film of surrogate organic aerosols (OA) species over a troposphere-relevant temperature range. The underlying hypothesis is that the temperature induced change in phase state of the studied chemicals will impact the uptake kinetics of atmospheric oxidants onto OA and consequentially the degradation rate of OA. This was clearly*

[Figure]

*shown from the measurement of uptake coefficient that took place in a temperature-controlled flow reactor with inner wall coated with the studied OA surrogate species. The phase state changes were supported by poke-flow experiments. The measured uptake coefficients onto the thin film were extended to estimate the temperature dependent particle degradation fraction (DF) for studied species/binary mixture as 200 nm particles. Atmospheric implications were drawn that the lifetimes of OA are expected to be longer against heterogeneous oxidation due to a liquid-to-solid or semi-solid to solid phase transition that slows down the reactions. The overall experimentations were well thought and designed to minimize effects that can impact the accurate determination of reaction kinetics. The results were presented in high quality and discussed in detail with appropriate reference to the literature for data interpretation and comparison. Some specific comments provided below:*

*1. Could authors give more background to justify the choice of canola oil, levoglucosan and glucose as OA surrogates and the choice of paired oxidant as well?*

Our choices of OA surrogate compounds were motivated by the fact of examining compounds that are representative of OA and for which the phase state properties are known. The choices of gas-phase radicals and oxidants include typical species of the polluted and background atmosphere to reflect oxidation processes close to the source region and during atmospheric transport, where $NO_3$ is representative of nighttime oxidation processes and OH and $O_3$ of daytime processes.

Canola oil (CA) is a mixture of multiple saturated and unsaturated fatty acids, however, dominated by unsaturated oleic and linoleic acid. Those species are prevalent components of both marine and terrestrial organic aerosol (Limbeck et al. 1999, Schauer et al. 2002, Kawamura et al. 2003) and constitute a major anthropogenic source of OA in urban environments from cooking (Rogge et al. 1991, Zahardis et al. 2007, Allan et al. 2010, Liu et al. 2017). Daytime photochemical aging of these compounds is simulated by studying the reaction with $O_3$ and OH oxidants.

Levoglucosan is applied in this study to reflect OA from biomass burning stemming from the pyrolysis of cellulose and hemicellulose products (Schauer et al. 2001, Iinuma et al. 2007). This reflects OA from natural sources such as forest fires and anthropogenic sources such as from heating and cooking. Sugars, such as glucose (GLU), can serve as tracers to characterize and apportion primary biogenic organic aerosols (PBOAs) (Zhu et al. 2015, Samaké et al. 2019). Previous studies, including our own, have indicated very slow oxidation reaction kinetics against $O_3$ (Knopf et al. 2011). To reflect urban day- and nighttime oxidation of these OA and its chemical transformation during long-range transport, we investigated the multiphase chemical kinetics involving $NO_3$ and OH radicals reflecting nighttime and daytime chemistry, respectively.

We will add the following sentences to the manuscript:

Line 105: "Canola oil (CA) is a mixture of multiple saturated and unsaturated fatty acids, however, dominated by unsaturated oleic and linoleic acid, and serves as OA surrogate for both marine and terrestrial organic aerosol including anthropogenic emissions like cooking (Rogge et al. 1991, Limbeck et al. 1999, Schauer et al. 2002, Kawamura et al. 2003, Liu et al. 2017). Levoglucosan (LEV) serves as a surrogate for biomass burning aerosol (BBA) including natural sources such as forest fires and anthropogenic sources such as heating and cooking BBA (Schauer et al. 2001, Iinuma et al. 2007). Glucose (GLU) can serve as a tracer to characterize and apportion primary biogenic organic aerosols (PBOAs) (Zhu et al. 2015, Samaké et al. 2019) and. Xylitol and 1,2,6-hexanetriol are used in compounds mixtures to modify the glass transition temperature, thus, condensed-phase viscosity. $O_3$ and OH radicals reflect typical gas oxidants present in photochemically active regions and during long-range transport. $NO_3$ radicals reflect an effective oxidant participating in the nighttime chemistry of typically polluted regions (Finlayson-Pitts et al. 1999, Brown et al. 2006)."

*2. There are no clear indications on the morphologies of films before being poked. Please consider adding these images to help interpret the images including those in the SI. Are the films for all individual species and mixtures supposed to be transparent*

*before poking?*

In general, all examined films were transparent and, considering a magnification of 230, appeared to have a smooth surface. When solidifying the film substrate, we try to avoid crystallization, which would result in a poly-crystalline surface distinctly different from film images presented here. The look of the images depends on the focal point and the thickness of the substrate. We focused on the film surface (except Fig. S4, image 2) to monitor the flow of the substrate to estimate phase state. When the film is thicker than the focal depth or the focal point is further away from the film surface, the substrate holder (i.e., the surface of the silver block in the temperature-controlled cooling stage) is barely- or non-visible, e.g., Fig. 3. If the film is thinner or the focal point is further within the substrate, you may see structures which stem from scratches in the sample holder, e.g., Figs. S2 and S3. (Recall, film is on thin glass slide on top of sample holder). Hence, the morphology of the substrate before poking is the area of the substrate not affected by the poking and potential shattering. For example, in the image of Fig. 3A at 10s, the area around the dent reflects the film surface properties as imaged by the microscope. The slightly darker shades within the imaged area do not represent any surface morphology but the unfocused scratches from the silver substrate holder beneath the film. Same with figures in supplement, however, in some of those cases, one can see the scratches in the sample holder.

To clarify this issue, we add the following sentences:

Line 187: "Images of the film substrate were recorded at 230 magnification and the depth of field of the 10X objective is 15.9 $\mu$m."

Line 192: "Images were recorded with the film surface in focus to monitor the flow of the organic substrate and to probe film viscosity. Under the applied resolution, all film substrates appeared to be smooth and transparent."

Line 208: "The area around the dent reflects the smooth morphology of the film substrate before poking it. The slightly darker shades within the imaged area do not rep-

resent any morphology features but the unfocused scratches from the silver substrate holder beneath the film."

Supplement, line 79: "Structures in Figs. S2-S7 that are not in focus are from minor superficial scratches of the silver sample holder within the temperature-controlled cooling stage."

*3. Line 224, would the $T_g^{pred}$ agree better with $T_g$ exp if 10%-16% residual water was accounted for in the calculation? This will help to support the argument here.*

If 10%-16% residual water is considered, using Gordon-Taylor equation, the predicted $T_g^{(pred,water)}$ would be lower than our experimentally determined $T_g^{exp}$. Comparing values of $T_g^{(pred,water)}$ and $T_g^{exp}$ yields for LEV 210 K and 243 K, for GLU 241 K and 273 K, for LEV/XYL 207 K and 238 K, and for GLU/HEX 216 K and 248 K, respectively. Hence, about 30 K in difference would be observed. This prediction assumes that the residual water is homogeneously distributed within the substrate. However, we suggest that this discrepancy is due to our substrate film preparation process where under slow drying (evaporation), the outermost layers of the film contain less water than the deeper layers. The outermost layers adjust quicker to the targeted phase state, potentially forming a core-shell configuration where the outer shell can display higher viscosity than the core/bulk. The focus in our poke-flow experiments is solely on the substrate surface, thus monitoring the substrate flow that is governed by the film viscosity closest representing the desired conditions. This effect will not impact the interpretation of our findings since, even in the case of a liquid substrate, the reacto-diffusive length is on the nanometer scale (George et al. 2010, Shiraiwa et al. 2011, Slade et al. 2014, Lee et al. 2016), much shorter than the overall depth of the substrate.

We add the following information to the text:

Line 224: "If 10%-16% residual water is considered while using Gordon-Taylor equation, the predicted $T_g$ would be roughly 30 K lower than our experimentally determined $T_g^{exp}$, assuming the residual water is homogeneously distributed in the film. This discrepancy is very likely due to our substrate film preparation process where under slow drying (evaporation), the outermost layers of the film contain less water than the deeper layers. Thus, the substrate surface represents closer the experimental conditions. Furthermore, the microscope focus in our poke-flow experiments is on the substrate surface, thereby monitoring the substrate morphology that is governed by the film viscosity closest representing the desired conditions."

*4. In Section "Atmospheric implications" and "Conclusions", author should probably note that when coupled with higher ambient relative humidity the estimated temperature dependent DF using uptake coefficient derived from dry experiments may be underestimated for hygroscopic species like levoglucosan and glucose. Some discussions in this regard and directions for future studies could be added.*

The reviewer is correct. We add to the Atmospheric Implications section: Line 466: "The presence of water vapor will significantly impact the phase state, in particular, of hygroscopic species such as LEV and GLU (Zobrist et al. 2008, Mikhailov et al. 2009, Koop et al. 2011), where increasing humidity yields lower condensed-phase viscosity and, in turn, faster reaction kinetics (Slade et al. 2014, Davies et al. 2015, Slade et al. 2017). Neglecting this effect will lead to an underestimation of DF."

For the Conclusions section, we will add:

Line 495: "This study did not address the role of water vapor acting as a plasticizer concurrent to phase changes induced by temperature changes (Zobrist et al. 2008, Mikhailov et al. 2009, Koop et al. 2011). The role of humidity on amorphous phase state and resulting multiphase kinetics has been studied at room temperature (Shiraiwa et al. 2011, Slade et al. 2014, Davies et al. 2015, Li et al. 2018). Most of these studies suggest that increasing humidity leads to faster reactive uptake kinetics. However, at lower temperatures, diffusivity is slower leading to kinetically hindered adjustments of the condensed-phase state (Wang et al. 2012, Berkemeier et al. 2014, Charnawskas et al. 2017, Knopf et al. 2018). Future experimental studies should focus on how the

coupled effects of ambient temperature and humidity on the amorphous phase state of OA particles modulate the multiphase oxidation kinetics.

*Technical Notes: 1. Font types of header and numbers in Table S4-Table S6 are not consistent.*

Corrected.

References

Allan, J. D., Williams, P. I., Morgan, W. T., Martin, C. L., Flynn, M. J., Lee, J., Nemitz, E., Phillips, G. J., Gallagher, M. W., and Coe, H.: Contributions from transport, solid fuel burning and cooking to primary organic aerosols in two UK cities, Atmos. Chem. Phys., 10, 647-668, http://doi.org/10.5194/acp-10-647-2010, 2010.

[revised manuscript text omitted]

---

## Author Comment (AC2) · 23 Apr 2020

We thank the reviewer for taking the time to carefully evaluate our manuscript and for the constructive comments. We feel that the comments improved the presentation of our study. Following the reviewer's suggestions, we will revise our manuscript correspondingly. Listed below are our point-by-point responses given in normal font and the reviewer's comments are given in italic font.

**Summary**

*The manuscript by Li et al. used both experimental techniques and different mode simulations (including a viscosity model and a reaction kinetics model) to systematically understand the influence of temperature on the multiphase reactive process of $O_3$, $NO_3$, and OH on levoglucosan, xylitol, glucose, 1,2,3-hexanetriol, and canola oil mixtures. The authors carefully designed a flow tube reactor to measure the reactive uptake coefficient, γ, of these three gas phase species with organic thin films from 213 to 293 K. The results suggest that the phase state of the organic compounds at cooler temperatures in the upper troposphere could heavily impact the multiphase reactions and the lifetime of the organic aerosols.*

*The experimental section and the kinetic model sections of the manuscript are well done. The results are very important to help understand the multiphase process under low temperature regimes. The interpretation of the experimental data is a little bit unclear and overstated and should be corrected in the revision. With the following points being addressed, this manuscript is suitable to be published in Atmospheric Chemistry and Physics.*

We thank the reviewer for the general positive evaluation.

**Major Comments**

*1. The author measured the reactive uptake coefficients of several species for residence times between 2 ms to 100 ms. Is the residence time safe to extrapolate the experimentally derived reactive uptake coefficients to ambient conditions where the residence time of the particles can be much longer? Shiraiwa et al. (2011) shows that the reactive uptake coefficients of ozone can vary significantly depending on the residence time. Can the author discuss about the potentially effects of short residence times in the manuscript?*

We believe there is a misunderstanding between the residence time in the flow reactor (i.e., reaction time) and the oxidant exposure. For example, at a fixed injector position, we can expose a substrate film for an hour, but the residence time of the gas oxidant is still in milliseconds. Hence, the interpretation comparing residence time with exposure time discussed in Shiraiwa et al. (2011) is misleading. We have previously discussed the relevance of the exposure time on the reaction mechanisms by applying a kinetic flux model (Arangio et al., 2015;Shiraiwa et al., 2012).

The concentrations of gas-phase oxidants applied in this study are around $10^{12}$, $10^{10}$ and $10^8$ molecules $cm^{-3}$ for $O_3$, $NO_3$, and OH, respectively. This corresponds to 40, 0.4 and 0.004 ppb at 293 K and 1 atm, which are representative of or close to typical atmospheric concentrations as discussed in section 2.2 entitled "Oxidant formation, detection and flow conditions". As outlined in the Supplement, application of these concentrations does not lead to a surface saturation effect on the derived kinetics. However, as outlined in Fig. 10, significant surface oxidation of OA can occur over typical exposure time periods in the atmosphere.

In the case of OH oxidation, our analysis supports that the oxidation reaction with a semi-solid or solid organic substrate is confined to the surface under applied OH concentrations for long time periods (Arangio et al., 2015). Related studies also indicate that, in the case of a liquid substrate, the reacto-diffusive length is on the order of nanometers (Slade and Knopf, 2014;George and Abbatt, 2010;Lee and Wilson, 2016;Shiraiwa et al., 2011). In the case of $NO_3$ oxidation of a solid substrate, we cannot rule out that under longer exposure as encountered in the atmosphere, bulk reaction processes may also play a role, thereby changing the reactivity as we have shown previously (Shiraiwa et al., 2012). In the case of $O_3$ oxidation of liquid and solid organic substrates, the results by Shiraiwa et al. (2011) indicate that the reactivity of $O_3$ oxidation decreases with exposure time. Therefore, in the cases of $NO_3$ and $O_3$ oxidation, we probably underestimate the chemical lifetime.

We add to the section "Atmospheric Implications"
Line 456: For OH oxidation, previous studies indicate that the oxidation reaction is confined to near the surface of a liquid or solid organic substrate, even for longer OH exposure periods at lower OH concentrations (Slade and Knopf, 2014;George and Abbatt, 2010;Lee and Wilson, 2016;Shiraiwa et al., 2011). However, whether bulk processes may significantly change the reactivity under long oxidant exposure as encountered in the atmosphere still needs to be examined. In the case of $O_3$ and $NO_3$ oxidation of organic substrates, the results by Shiraiwa et al. (2011) and Shiraiwa et al. (2012) indicate that the reactive uptake coefficients decrease with increasing exposure. Therefore, we probably underestimate the chemical lifetime.

*2. The trends of the γ values as a function of temperature are really interesting and it shows strong heterogeneity among different species. The authors seem to imply in the manuscript that phase state change is the dominant reason for γ values to change (line 18, line 484-485, line 496-497). However, Shiraiwa et al. (2011&2012) demonstrate that if the organic species change their phase state from liquid to glass, the mass accommodation coefficients will change by orders of magnitude rather than just 34 times. Could the authors comment on the discrepancy between previous modeling results showing the uptake could change by orders of magnitude vs. the current experimental data showing the γ values only change up to a factor of 34?*

With respect to the surface accommodation coefficient, Shiraiwa and Seinfeld (2012) present a theoretical sensitivity study on how this coefficient impacts the equilibration timescale of SOA gas-particle partitioning. Therefore, this research is not directly relevant for interpretation of our experimentally derived reactive uptake coefficients. Shiraiwa et al. (2011) discriminates between the surface and bulk mass accommodation coefficient being 1 and $10^{-5}$ for the entirety of their experiments including the presence of different substrate phase states. Hence, there is no change in their accommodation coefficients but after a certain exposure time period the reactive uptake coefficient is lower than the bulk mass accommodation coefficient. We cannot derive the mass accommodation coefficients directly from the experimental data, but it is reasonable to assume that the surface mass accommodation coefficient is around 1 as well. However, since our reactive uptake coefficients are greater than in the study by Shiraiwa et al. (2011), the bulk mass accommodation coefficient is likely also larger.

Regarding to the variation of the reactive uptake coefficient with the phase change, we observed about one order in magnitude change in the reactivity towards $O_3$ when the substrate phase state changes from liquid to solid. This is in agreement with the observations by Shiraiwa et al. (2011), Steimer et al. (2015), and Berkemeier et al. (2016), besides the direct comparison with the study by de Gouw and Lovejoy (1998). Overall, we observe very similar changes in the multiphase chemical kinetics.

*With carefully designed experiments, I do believe the γ values the authors presented are accurate, but I think the data interpretation part is a little bit overstated to conclude phase state is the main reason for γ to change. The authors need to discuss more why γ values did not decrease as fast as previous modeling results (Shiraiwa et al., 2011&2012) shown when the viscosity changes. For instance, could the desorption lifetime (line 331) help increase γ values as temperature decreases? I think it is important to discuss these aspects in the paper to show that maybe both phase state and the desorption kinetics play important, but counter roles, with the role of the phase state dominating the γ value during this process.*

As discussed above, the observed change in $O_3$ reactivity with phase state is similar to previous studies. The role of the desorption lifetime in multiphase chemical kinetics and its dependence is yet not resolved. Desorption lifetime is assumed as a constant on the order of nanoseconds in most recent modeling studies (Berkemeier et al., 2016;Arangio et al., 2015;Shiraiwa et al., 2012;Shiraiwa et al., 2011). For OH oxidation, we have discussed the importance of the desorption lifetime in Arangio et al. (2015). In this manuscript that focuses on the experimental results, we do not want to add too much speculation. Application of a kinetic flux model as in our previous studies (Shiraiwa et al., 2012;Arangio et al., 2015;Springmann et al., 2009;Kaiser et al., 2011) could assist in furthering this discussion in the future. We have included the importance of desorption lifetime and its potential counter role at several places in the manuscript including the conclusions, e.g., lines 321, 331 and 489. Although we cannot quantify the effect of the desorption lifetime on the observed kinetics, the greatest changes in reactivity commence when the substrate undergoes a phase transition.

*3. The authors performed the poke-flow technique that was used in other papers to determine the glass transition temperatures of organic species. However, past literature on poke-flow technique were mostly used to determine the viscosity of the organic species and glass shattering phenomenon can indicate the viscosity is anywhere >$10^{12}$ Pa s (Lindsay et al. 2013), but does not necessarily mean the viscosity is exactly near $10^{12}$ Pa s when the compound is right around glass transition. The accuracy of the glass transition temperature derived from this technique will heavily impact data interpretation in Fig. 7-9. Given that the $T_g$ values shown in Table 1 do not match the $T_g$ values from literature (and the author also mentioned these compounds may have water in them, so it is difficult to compare the $T_g$ values with pure compounds from the literature), can the author provide any evidence to consistently show that the poke-flow technique can be used to derived accurate $T_g$ values? Given there hasn't been studies before using observation on glass shattering to solely determine the glass transition temperature, maybe the author can cross compare with previous data to validate this technique (such as Koop et al. 2010, Dette et al. 2015).*

First, we want to emphasize that we clearly mentioned in the manuscript at line 182 and 185, our poke-flow technique is a semi-quantitative method to probe the phase transition of applied substrate films within the investigated temperature range. We do not know any study that has probed and controlled such a large organic substrate for oxidation kinetics experiments. The poke-flow technique gives us the means to ascertain or rule out specific phase states. We do not claim that we can measure accurately the continuous viscosity change of applied substrates. Although, when looking at revised Table 1 (given below) that includes the uncertainty of literature values of $T_g$ and $T_g$ predictions, we achieve agreement or close agreement with those values.

We agree with the reviewer that the glass shattering phenomenon only indicates that the viscosity is >$10^{12}$ Pa s. To minimize this uncertainty, we measure $T_g$ as the highest temperature when shattering occurs. For example, if the GLU substrate films were poked at 276 K and 273 K, and the shattering phenomenon did not occur at 276 K but at 273 K, we know that the viscosity is smaller than $10^{12}$ Pa s at 276 K but larger than $10^{12}$ Pa s at 273 K. Of course, if we poke the same film at 270 K, the shattering occurs, and the viscosity

will probably be larger than $10^{12}$ Pa s at this temperature. Therefore, we assign the shattering of the substrate film a viscosity of $10^{12}$ Pa s at the highest determined temperature. We will make an additional remark in the manuscript to point out this uncertainty:

Line 219: "Viscosity greater than $10^{12}$ Pa s indicates the presence of a glassy phase. Thus, we ascribe the observed shattering of the substrate film a viscosity of $10^{12}$ Pa s, although the exact value cannot be assessed with the poke-flow technique."

We revised Table 1 and changed the main text accordingly:
We changed line 220:

" $T_g^{\mathrm{pred}}$ is derived from the Gordon Taylor equation (Gordon and Taylor, 2007) assuming no residual water in the organic mixture. Since $T_g$ determination depends on cooling or drying rates, we do not expect agreement with literature values or predicted values. For most investigated substrates, our estimated $T_g^{\mathrm{exp}}$ is lower than either $T_g^{\mathrm{lit}}$ or $T_g^{\mathrm{pred}}$. This is consistent with the notion that our substrates contain some residual water that lowers $T_g$."
To

" $T_g^{\mathrm{pred}}$ for substrate mixtures is derived from the Gordon Taylor equation (Gordon and Taylor, 2007) assuming literature $T_g$ values and no residual water present. The uncertainties in $T_g^{\mathrm{pred}}$ values are derived by Gaussian error propagation. Except for the GLU substrate film, we achieve agreement or close agreement with either $T_g^{\mathrm{lit}}$ or $T_g^{\mathrm{pred}}$. Since $T_g$ determination depends on cooling or drying rates, we do not expect agreement with literature values or predicted values."

To address the issue of residual water, this is followed by text in response to reviewer #1:

"In addition, if 10%-16% residual water is considered while using Gordon-Taylor equation, $T_g^{\mathrm{pred}}$ would be roughly 30 K lower than $T_g^{\mathrm{exp}}$, assuming the residual water is homogeneously distributed in the film. This discrepancy is very likely due to our substrate film preparation process where under slow drying (evaporation), the outermost layers of the film contain less water than the deeper layers. Thus, the substrate surface represents closer the experimental conditions. Furthermore, the microscope focus in our poke-flow experiments is on the substrate surface, thereby monitoring the substrate morphology that is governed by the film viscosity closest representing the desired conditions."

The new Table 1 reads:

**Table 1.** Estimated $T_g$ of the applied substrate films. $T_g^{\mathrm{exp}}$ is the measured $T_g$ using the poke-flow technique. $T_g^{\mathrm{lit}}$ is the literature reported $T_g$. $T_g^{\mathrm{pred}}$ is predicted $T_g$ for mixtures by Gordon-Taylor equation using $T_g^{\mathrm{lit}}$ and $k_{GT}$. $k_{GT}$ is solute specific constant in the Gordon-Taylor equation. $D$ is fragility.

| Substrate | $T_g^{\mathrm{exp}}$ / K | $T_g^{\mathrm{pred}}$ / K | $T_g^{\mathrm{lit}}$ / K | $k_{GT}$ | $D$ |
|---|---|---|---|---|---|
| Levoglucosan (LEV) | 243 ± 4 | | 248 ± 2 [a,h,g] | 3.26[a] | 14.1[b] |
| Xylitol (XYL) | | | 249 ± 7 [f,g,i] | 2.1[c] | 8.65[b] |
| LEV/XYL | 238 ± 3 | 249 ± 5 | | | 11.3[d] |
| Glucose (GLU) | 273 ± 3 | | 305 ± 13 [e,g,j] | 3.95[e] | 12.1[b] |

| | | | | | |
|---|---|---|---|---|---|
| 1,2,6-Hexanetriol (HEX) | | | $204 \pm 6$ [f,g,i] | $0.88$ [e] | $13.16$ [f] |
| GLU/HEX | $248 \pm 3$ | $252 \pm 7$ | | | $12.3$ [d] |

[a] Lienhard et al. (2012). [b] DeRieux et al. (2018). [c] Elamin et al. (2012). [d] Interpolated values based on mass fraction. [e] Zobrist et al. (2008). [f] Nakanishi et al. (2011). [g] Rothfuss (2019). [h] Tombari and Johari (2015). [i] Dorfmüller et al. (1979). [j] Diogo and Ramos (2008).

*4. What is the relative humidity of the flow tube when the experiments were performed? Given the author performed the flow experiments under a variety of low temperatures, even small partial pressure of water could potential lead to relatively high humidity and alter the phase state of the organic film and increase the error bar range of the estimated viscosity. Another related question is that the poke flow measurement also was performed under "sealed room air" (line 190). The water vapor in the room air may lead to near high relative humidity at cooler temperatures. How would the authors ensure the RH levels in the flow tube and the poke flow cooling state are the same and don't lead to errors in estimating the phase state?*

The reviewer is correct that miniscule amounts of water vapor could lead to high humidity levels in the flow reactor at low temperatures. All experiments were conducted under dry conditions. Extra care has been taken as described in our experimental section (section 2.2, line 141). We apply a carbon filter (Supelco, Superlcarb HC) and a Drierite cold trap cooled with liquid nitrogen or an ethanol/dry ice mixture to purify all the ultra-high purity (UHP) gases, such as $N_2$, He and $O_2$, which yields insignificant water partial pressures. The system is a vacuum system and thus leakage of room air resulting in water vapor contamination can be excluded.

Also, for the poke flow experiments, as discussed in section 2.3 line 190, the organic substrate film placed within the cooling stage is sealed against room air during the entire poking process, residing in a dry $N_2$ atmosphere. Only pure $N_2$ gas from a liquid nitrogen container is purged into the cooling stage to keep a positive pressure environment. Therefore, this measure excludes contamination of water vapor from room air.
We will change the sentence on line 190
"In the cooling stage the organic film sample is sealed against room air during the entire poking process."
To
"In the cooling stage, the organic film sample is sealed against room air and resides in an atmosphere of dry $N_2$ at positive pressure during the entire poking process."

*5. The author used "slight changes in viscosity" (line 330) and "small changes in viscosity" (line 491) multiple times in the manuscript to describe viscosity change as temperature drops. However, the changes in viscosity during semi-solid phase state spans over several orders of magnitude (Figure 4) and I would not consider these changes small. In fact, these changes could be even larger than the phase state transition from liquid to semi-solid. This leads to my question when viscosity changes a few orders of magnitude from liquid to semi-solid, it seems the uptake coefficients are still the same.*

We agree with this statement of the reviewer and we will re-phrase these statements. We want to clarify how we interpret the data regarding the uptake coefficients change little although viscosity changes by several orders of magnitude. Please recall our point above that the poke-flow technique is a semi-quantitative approach. However, we can, rather accurately, determine the glassy and liquid phase state

of the substrate but not the changes in between these extreme points. We observe that the greatest changes in the reactive uptake occur when the substrate phase changes from solid or semi-solid to liquid. This is evident for $O_3$ and OH reacting with canola oil and for $NO_3$ reacting with the LEV/XYL mixture and to a lesser extent for OH reacting with the GLU/HEX mixture. For those observations, without a detailed model, we cannot assign a definitive change in viscosity. Figure 4 serves as guidance only and will need further investigation. We observe that the reactive uptake does not change significantly when the substrate is in the solid and semi-solid phase regime. This is the case for $NO_3$ uptake by LEV and OH uptake by GLU. Literature data depicted in Fig. 4 demonstrates that LEV and GLU substrate films indeed do not transition into the liquid phase in the experimentally probed temperature range. Hence, although viscosity changes by several orders of magnitude in the semi-solid regime, the self-diffusion of organic molecules and characteristic mixing time scales are, even at lowest viscosity, similar and longer than typical experimental times, thereby contributing to a lesser impact on the reactive uptake. Therefore, the reactivity changes less in this regime than it does from a semi-solid to liquid phase.

We will change the sentence on line 330:
"The slight increase of γ with temperature may due to the slight change in substrate viscosity (see Fig. 4), the counteracting effect of the temperature dependence of the reaction rate (Atkinson et al., 2006) and the desorption lifetime (Pöschl et al. 2007)."
to
"The slight increase of γ with temperature may be due to the combined effects of changes in substrate viscosity (see Fig. 4), the counteracting effect of the temperature dependence of the reaction rate (Atkinson et al., 2006) and the desorption lifetime (Pöschl et al., 2007). Assessment of these various impacts on the observed uptake kinetics necessitates an in-depth analysis, e.g., by application of kinetic flux modeling (Shiraiwa et al., 2012;Arangio et al., 2015)."

We change the sentence on line 491:
"Small changes in viscosity with temperature may also play a role in the overall heterogeneous kinetics when the substrate is in a semi-solid phase state."
To
"Changes in substrate viscosity with temperature may also play a role in the overall heterogeneous kinetics when the substrate is in a semi-solid phase state."

*6. The logical deductions of how phase state alters γ values are a bit confusing and somewhat contradictory in multiple parts of the manuscript. For instance, line 339 shows the largest change of γ is observed around the expected Tg for LEV/XYL film, showing a transition from semi-solid to glass regime has the maximum influence of γ. However, in the next sentence, the author made a contradictory comment (line 340-341), saying "significant change in $NO_3$ uptake reactivity in the investigated temperature range can be attributed to the transition of a liquid to a highly viscous or solid substrate film". The similar statement appeared in line 351-352 saying that "the lower sensitivity of γ on temperature may be due to the generally higher viscosity" of the compound studied. Again, in line 358-400 the authors state that the γ is less sensitive to temperature change when canola is in semi-solid/glass phase states. If the γ change is less obvious, then why would the author observe the largest change of γ around Tg for LEV/XYL film?*

We agree with the reviewer's assessment that our statements can cause confusion. In general, we meant that the largest changes in γ are observed above the expected $T_g$ (e.g., line 339). This may have caused confusion at subsequent instances. We change those sentences accordingly:

Line 339: "The largest change of γ is observed around the expected $T_g$ (Fig. 8b). Therefore, the significant change in $NO_3$ uptake reactivity in the investigated temperature range can be attributed to the transition of a liquid to a highly viscous or solid substrate film."
Is changed to
"The largest change of γ is observed above the expected $T_g$ (Fig. 8b). Therefore, the significant change in $NO_3$ uptake reactivity in the investigated temperature range can be attributed to the transition of a solid or highly viscous to a liquid substrate film."

Line 338: To avoid repetition, we delete "The observed change in uptake kinetics is attributed to the phase change of the film substrate. "

Line 351: "Therefore, the lower sensitivity of γ on temperature may be due to the generally higher viscosity compared to a glycerol substrate."
Is changed to
"We hypothesize that the lesser change in γ with temperature for the LEV/XYL substrate compared to a GLY substrate (Gross et al. 2009), may be due to the viscosity difference between both substrates."

Line 349: To avoid repetition, we delete "The reason for this difference may lie in the higher viscosity of the LEV/XYL substrate within this temperature range. "

Line 390: "The major change in γ matches the expected phase change of the GLU/HEX mixture."
Is changed to
"The major change in γ occurs above the expected $T_g$ of the GLU/HEX mixture."

Line 396: "γ increases by a factor of ~2.4 between 273 K and 293 K, where the viscosity of canola oil changes from 0.185 to 0.079 Pa s (Fasina et al., 2006)."
Is changed to
"Above the melting point temperature, γ increases by a factor of ~2.4 between 273 K and 293 K, where the viscosity of canola oil changes from 0.185 to 0.079 Pa s (Fasina et al., 2006)."

*The authors also compared the γ change with the same amount of temperature change (25 K) at higher temperature regimes (liquid) and lower temperature regimes (semi-solid/solid) in the manuscript. Results show that that γ changes more significantly at higher temperature regimes (liquid). My question is, would the statement still be correct when comparing viscosity change instead of temperature change? I.e., will γ still change more significantly when viscosity changes the same orders of magnitude from liquid to semi-solid vs. semi-solid to glass?*

We could plot reactive uptake coefficients as a function of viscosity using data of Fig. 4. However, the viscosity data represents only estimates and as such we refrain from this exercise. In general, we would expect a greater sensitivity of γ on temperature and thus viscosity in the liquid phase regime. The lower the viscosity, the faster the diffusivity and in turn more reactive sites are available. In the semi-solid and solid phase regime, the sensitivity of γ is likely lower, since if the viscosity is sufficiently low and characteristic diffusion time scales are longer than experimental time scales, no difference in reactivity due to phase or viscosity changes may be observed.

We add the following sentences to the section "Conclusions":

Line 478: "Furthermore, once in the liquid phase, as temperature increases viscosity decreases and diffusivity increases, leading to potentially strong increases in reactivity."

Line 481: "Although viscosity in the semi-solid phase regime can change substantially with temperature, viscosity can still be too high to allow for significant bulk processes to play a role."

*7. The reactive uptake data as a function of temperature from this manuscript is very insightful since there have not been such data available at low temperature. Results also show that change of γ as a function of temperature is drastically different when it comes to different organic species. For instance, OH oxidation of GLU/HEX mix and NO3 oxidation of LEV is not sensitive to temperature while OH and O3 oxidation oxidations of canola oil are more sensitive to temperature change. Can the author discuss further in the manuscript why different species exhibit difference in γ value changes? Given the strong heterogeneity of how the γ values of different compounds respond to temperature change, can the author still make the conclusions that "chemical reactivity of organic matter towards atmospheric oxidants can vary significantly in response to ambient temperature", as the ambient OA may have a different response of γ towards temperature than the organic surrogate compounds used in this study?*

In response to "*. Can the author discuss further in the manuscript why different species exhibit difference in γ value changes?".* We have not discussed this point in our conclusion section. However, in the discussion section, we explored the differences in reactivity of the examined gas-substrate systems. Based on gas-phase reaction kinetics, we expect different reaction pathways for the systems studied. For example, $O_3$ will preferably react with a double bond, and OH and $NO_3$ abstract a hydrogen. As deduced from gas-phase chemistry, different reaction systems will show different reaction rates and different dependencies with temperature. See for example the decrease in reactivity with increasing temperature for the reaction of glucose with OH. We will add these points in the atmospheric implication section. In addition to this, we have to deal with the effect of phase state on reactivity (analogously to pressure dependence of gas-phase kinetics). For each substrate, the viscosity variation with temperature change is different, and thus the diffusion coefficients of organic molecules and gas oxidants can be quite different.

Regarding the second point of the reviewer. Yes, it still holds that "*chemical reactivity of organic matter towards atmospheric oxidants can vary significantly in response to ambient temperature*". If we assume multiphase and multicomponent particles with complex composition and morphology (Laskin et al., 2016;Laskin et al., 2019), which can potentially lead to matrix effects and phase separation (Lignell et al., 2014;Lee and Wilson, 2016;Charnawskas et al., 2017;Bertram et al., 2011), clearly, the applied single and binary substrate films are insufficient to describe these scenarios. However, OA can only cycle between solid and liquid phase states, and the temperature and humidity under which these phase changes occur will depend on organic compounds and the presence of the plasticizer (i.e., water, sulfates, etc.) (Mikhailov et al., 2009;Virtanen et al., 2010;Koop et al., 2011;Reid et al., 2018). For example, Shiraiwa et al. (2017) estimates that the majority of SOA particles transition from liquid to solid with increasing altitude (and decreasing temperature) independent of the particles' chemical complexity. Our multiphase chemistry experiments cover a broad temperature range and corresponding range of phase states typically encountered in the atmosphere and thus can provide useful estimates and atmospheric implications. We agree with the reviewer and we will add the issue of heterogeneity to our conclusions section.

We add to the section "Atmospheric Implications":
Line 466:

"This discussion neglects the chemical complexity of ambient OA where different condensed-phase species can result in different reactivities and reaction pathways (Zhang et al., 2015;Surratt et al., 2010;Ziemann and Atkinson, 2012;Knopf et al., 2005;Davies and Wilson, 2015). Those in turn can change the multiphase kinetics and its dependency on temperature and particle phase state. Furthermore, heterogeneous particle composition and morphology can result in matrix effects or liquid-liquid phase separation, where, e.g., more reactive organic species are shielded by less reactive species (Lignell et al., 2014;Lee and Wilson, 2016;Charnawskas et al., 2017;Bertram et al., 2011). Those effects were not assessed in this study but necessitate additional experimental investigations.

In the section "Conclusions" we add:
Line 497:
"Ambient OA, however, display greater chemical and morphological complexity (Laskin et al., 2016;Laskin et al., 2019), and as such we expect varying multiphase reaction pathways having different reactivity towards atmospheric gas-phase oxidants which will translate into different reactivity dependencies on temperature and phase state. Despite this caveat, due to lower temperatures at higher altitudes, we can expect OA particles during transport in the free troposphere to have significantly longer lifetimes with respect to chemical degradation."

***Minor Comments***

*1. Line 287: The equation doesn't seem to be right. Why would the diffusion coefficient to the left side of the equation be in the numerator, but the diffusion coefficients to the right side of the equation be in the denominator? Could the authors please double check?*

Thank you for pointing this out. It is a typo; we forgot to take the "inverse" and is corrected as below:

$$D_X = \left( \frac{P_{He}}{D_{X-He}} + \frac{P_{O_2}}{D_{X-O_2}} \right)^{-1}$$

*2. The literature citation in the introduction part is good overall, but missed some recent important publications that are highly relevant to this study. Please include those citations for a more comprehensive introduction.*

The reviewer points out many nice and important publications to be cited in the introduction. We really appreciate these suggestions and will include some of those articles. However, there are several that, although important studies, we will not include. The reason is that we want to keep the focus on relevant studies that discuss the oxidation kinetics between the investigated gaseous oxidants (i.e., $O_3$, $NO_3$ and OH) and organic substrates. This already resulted in 137 references. To not further distract the reader, we reference studies on SOA formation, gas-particle partitioning, photochemical aging studies, or reactive uptake studies of other gas species only in a very limited manner. Some of the not directly relevant studies are covered by the referenced review articles.

*Line 40: please add the following two papers that are relevant. Murray, B. J., T. W. Wilson, S. Dobbie, Z. Cui, S. M. R. K. Al-Jumur, O. Mohler, M. Schnaiter, R. Wagner, S. Benz, M. Niemand, H. Saathoff, V. Ebert, S. Wagner and B. Karcher (2010). "Heterogeneous nucleation of ice particles on glassy aerosols under cirrus conditions." Nature Geosci 3(4): 233-237.*

We will add the cited Murray et al. (2010) study and the study by Wang et al. (2012) showing the ice nucleation ability of glassy SOA or SOA surrogate particles.

*Pöschl, U. and M. Shiraiwa (2015). "Multiphase Chemistry at the Atmosphere – Biosphere Interface Influencing Climate and Public Health in the Anthropocene." Chemical Reviews 115(10): 4440-4475.*

We have already cited this paper on line 31, line 53 and line 487 in the manuscript. We do not feel we need to add it to this list.

*Line 42: please add the following two papers that are relevant.*
*Renbaum-Wolff, L., J. W. Grayson, A. P. Bateman, M. Kuwata, M. Sellier, B. J. Murray, J. E. Shilling, S. T. Martin and A. K. Bertram (2013). "Viscosity of $\alpha$-pinene secondary organic material and implications for particle growth and reactivity." Proceedings of the National Academy of Sciences 110(20): 8014-8019.*

This paper has been cited in line 185. We will also add this reference on line 42.

*Kidd, C., V. Perraud, L. M. Wingen and B. J. Finlayson-Pitts (2014). "Integrating phase and composition of secondary organic aerosol from the ozonolysis of $\alpha$-pinene." Proceedings of the National Academy of Sciences 111(21): 7552-7557.*

We will add this reference.

*Line 44-45: please include the following two papers that are relevant.*
*Dette, H. P. and T. Koop (2015). "Glass Formation Processes in Mixed Inorganic/Organic Aerosol Particles." The Journal of Physical Chemistry A 119(19): 4552-4561.*

*Zhang, Y., L. Nichman, P. Spencer, J. I. Jung, A. Lee, B. K. Heffernan, A. Gold, Z. Zhang, Y. Chen, M. R. Canagaratna, J. T. Jayne, D. R. Worsnop, T. B. Onasch, J. D. Surratt, D. Chandler, P. Davidovits and C. E. Kolb (2019). "The Cooling Rate- and Volatility-Dependent Glass-Forming Properties of Organic Aerosols Measured by Broadband Dielectric Spectroscopy." Environmental Science & Technology 53(21): 12366-12378.*

We will include these two references.

*Line 47: please include the following three papers that are relevant*
*Rothfuss, N. E. and M. D. Petters (2016). "Coalescence-based assessment of aerosol phase state using dimers prepared through a dual-differential mobility analyzer technique." Aerosol Science and Technology 50(12): 1294-1305.*

*Zhang, Y., M. S. Sanchez, C. Douet, Y. Wang, A. P. Bateman, Z. Gong, M. Kuwata, L. Renbaum-Wolff, B. B. Sato, P. F. Liu, A. K. Bertram, F. M. Geiger and S. T. Martin (2015). "Changing shapes and implied viscosities of suspended submicron particles." Atmos. Chem. Phys. 15(14): 7819-7829.*

*Jarvinen, E., K. Ignatius, L. Nichman, T. B. Kristensen, C. Fuchs, C. R. Hoyle, N. Hoppel, J. C. Corbin, J. Craven, J. Duplissy, S. Ehrhart, I. El Haddad, C. Frege, H. Gordon, T. Jokinen, P. Kallinger, J. Kirkby, A. Kiselev, K. H. Naumann, T. Petaja, T. Pinterich, A. S. H. Prevot, H. Saathoff, T. Schiebel, K. Sengupta, M. Simon, J. G. Slowik, J. Trostl, A. Virtanen, P. Vochezer, S. Vogt, A. C. Wagner, R. Wagner, C. Williamson, P. M. Winkler,*

*C. Yan, U. Baltensperger, N. M. Donahue, R. C. Flagan, M. Gallagher, A. Hansel, M. Kulmala, F. Stratmann, D. R. Worsnop, O. Mohler, T. Leisner and M. Schnaiter (2016). "Observation of viscosity transition in $\alpha$-pinene secondary organic aerosol." Atmos. Chem. Phys. 16(7): 4423-4438.*

Here, we only cite studies to show, in general, that the amorphous phase state is modulated by temperature and humidity. There, is no need to cite the suggested very specific studies. In fact, we could just cite only Koop et al. (2011).  Hence, we will not include these references.

*Line 51: please add the following two relevant papers:*
*Shiraiwa, M. and J. H. Seinfeld (2012). "Equilibration timescale of atmospheric secondary organic aerosol partitioning." Geophysical Research Letters 39(24): L24801.*
*Renbaum-Wolff, L., J. W. Grayson, A. P. Bateman, M. Kuwata, M. Sellier, B. J. Murray, J. E. Shilling, S. T. Martin and A. K. Bertram (2013). "Viscosity of $\alpha$-pinene secondary organic material and implications for particle growth and reactivity." Proceedings of the National Academy of Sciences 110(20): 8014-8019.*

Shiraiwa and Seinfeld (2012) are referenced in the subsequent sentence. We will move the reference to this place. We will add Renbaum-Wolff et al. (2013).

*Line 53: please include the following two papers that are relevant:*
*Gaston, C. J., J. A. Thornton and N. L. Ng (2014). "Reactive uptake of N2O5 to internally mixed inorganic and organic particles: the role of organic carbon oxidation state and inferred organic phase separations." Atmos. Chem. Phys. 14(11): 5693-5707.*
*Zhang, Y., Y. Chen, A. T. Lambe, N. E. Olson, Z. Lei, R. L. Craig, Z. Zhang, A. Gold, T. B. Onasch, J. T. Jayne, D. R. Worsnop, C. J. Gaston, J. A. Thornton, W. Vizuete, A. P. Ault and J.D. Surratt (2018). "Effect of Aerosol-Phase State on Secondary Organic Aerosol Formation from the Reactive Uptake of Isoprene-Derived Epoxydiols (IEPOX)." Environmental Science & Technology Letters 5(3): 167-174.*

We will add those two references.

*Line 55: Please add the following paper that is relevant:*
*Riedel, T. P., Y.-H. Lin, S. H. Budisulistiorini, C. J. Gaston, J. A. Thornton, Z. Zhang, W. Vizuete, A. Gold and J. D. Surratt (2015). "Heterogeneous Reactions of Isoprene-Derived Epoxides: Reaction Probabilities and Molar Secondary Organic Aerosol Yield Estimates." Environmental Science & Technology Letters 2(2): 38-42.*

No further citation is need here. We just refer to the definition of the reactive uptake coefficient. Stephen Schwartz developed the mathematical groundwork for the resistor model and Pöschl et al. (2007) introduced the kinetic flux model approach which is consistent with the resistor model.

*Line 61: please include the following citations that are highly relevant to the paper:*
*Pajunoja, A., W. Hu, Y. J. Leong, N. F. Taylor, P. Miettinen, B. B. Palm, S. Mikkonen, D. R. Collins, J. L. Jimenez and A. Virtanen (2016). "Phase state of ambient aerosol linked with water uptake and chemical aging in the southeastern US." Atmos. Chem. Phys. 16(17): 11163-11176.*

We will add this reference.

*Gaston, C. J., J. A. Thornton and N. L. Ng (2014). "Reactive uptake of N2O5 to internally mixed inorganic and organic particles: the role of organic carbon oxidation state and inferred organic*

*phase separations." Atmos. Chem. Phys. 14(11): 5693-5707.*

Here, we want to place the focus on studies that specifically target multiphase oxidation reactions involving O₃, NO₃, and OH and discuss reaction rates. We cannot include all gas uptake experiments (please see UIPAC reviews on those, e.g., by Ammann et al. (2013) and Crowley et al. (2013) or reviews by Kolb et al. (2010), Davidovits et al. (2006), and other colleagues), gas-partitioning, SOA formation, or photochemical aging studies. Also, those studies are not directly comparable and relevant to the experiments presented here. For these reasons, we do not add this work here but in line 53.

*Houle, F. A., A. A. Wiegel and K. R. Wilson (2018). "Predicting Aerosol Reactivity Across Scales: from the Laboratory to the Atmosphere." Environmental Science & Technology 52(23):*
*13774-13781.*

Here we focus experimental studies involving RH effects on kinetics, and thus will not include this article at this place. We already cited this study on line 487 in the conclusion part while mentioning detailed modelling studies.

*Zhang, Y., Y. Chen, A. T. Lambe, N. E. Olson, Z. Lei, R. L. Craig, Z. Zhang, A. Gold, T. B. Onasch, J. T. Jayne, D. R. Worsnop, C. J. Gaston, J. A. Thornton, W. Vizuete, A. P. Ault and J. D. Surratt (2018). "Effect of Aerosol-Phase State on Secondary Organic Aerosol Formation from the Reactive Uptake of Isoprene-Derived Epoxydiols (IEPOX)." Environmental Science & Technology Letters 5(3): 167-174.*

As discussed above, we keep the main focus on determination of reaction kinetics and on the oxidants examined in this study. For these reasons, we do not add this work.

*Riva, M., et al. (2019). "Increasing Isoprene Epoxydiol-to-Inorganic Sulfate Aerosol (IEPOX:Sulfinorg) Ratio Results in Extensive Conversion of Inorganic Sulfate to Organosulfur Forms: Implications for Aerosol Physicochemical Properties." Environmental Science & Technology 53(15): 8682-8694.*

As discussed above, we keep the main focus on determination of reaction kinetics and on the oxidants examined in this study. For these reasons, we do not add this work.

*Zhang, Y., Y. Chen, Z. Lei, N. E. Olson, M. Riva, A. R. Koss, Z. Zhang, A. Gold, J. T. Jayne, D. R. Worsnop, T. B. Onasch, J. H. Kroll, B. J. Turpin, A. P. Ault and J. D. Surratt (2019). "Joint Impacts of Acidity and Viscosity on the Formation of Secondary Organic Aerosol from Isoprene Epoxydiols (IEPOX) in Phase Separated Particles." ACS Earth and Space Chemistry 3(12): 2646-2658.*

As discussed above, we keep the main focus on determination of reaction kinetics and on the oxidants examined in this study. For these reasons, we do not add this work.

*DeRieux, W.-S. W., P. S. J. Lakey, Y. Chu, C. K. Chan, H. S. Glicker, J. N. Smith, A. Zuend and M. Shiraiwa (2019). "Effects of Phase State and Phase Separation on Dimethylamine Uptake of Ammonium Sulfate and Ammonium Sulfate−Sucrose Mixed Particles." ACS Earth and Space Chemistry 3(7): 1268-1278.*

As discussed above, we keep the main focus on determination of reaction kinetics and on the oxidants examined in this study. For these reasons, we do not add this work.

*Line 64: please include the following citation that is relevant:*
*Arangio, A. M., J. H. Slade, T. Berkemeier, U. Pöschl, D. A. Knopf and M. Shiraiwa (2015). "Multiphase chemical kinetics of OH radical uptake by molecular organic markers of biomass burning aerosols: humidity and temperature dependence, surface reaction, and bulk diffusion." The Journal of Physical Chemistry A 119(19): 4533-4544.*

We will add this reference.

*Line 70-73: there are other works that also analyzed the effects of phase states on OH oxidation that should be included:*
*Houle, F. A., A. A. Wiegel and K. R. Wilson (2018a). "Predicting Aerosol Reactivity Across Scales: from the Laboratory to the Atmosphere." Environmental Science & Technology 52(23): 13774-13781.*
*Houle, F. A., A. A. Wiegel and K. R. Wilson (2018b). "Changes in Reactivity as Chemistry Becomes Confined to an Interface. The Case of Free Radical Oxidation of C30H62 Alkane by OH." The Journal of Physical Chemistry Letters 9(5): 1053-1057.*

In this text section we only reference studies that investigate the effect of RH on phase state and reaction kinetics. The cited articles do not discuss this effect.
However, as discussed above, we already cited Houle et al. (2018a) in our conclusion section. We will also cite Houle et al. (2018b) on line 52 in the introduction section.

*3. The measured Tg values of a few species listed in Table 1 are lower than the value measured by other literature. The author explains that was due to water remaining in the organic compound which lowers the glass transition temperature. If that is the case, it is a bit misleading to still associate the experimentally derived glass transition temperatures with the name of the pure compounds. Maybe the author should add information to highlight that the experimental derived Tg values in Table 1 refers to the mixture of water and organic species.*

Please see our response above, revised table, and response to reviewer #1.

*4. n-Hexadecane in Table 2 was never mentioned in the main text. Please call out the name. Also, in Table2, a comma should be added between reference number and the γ values*

Thank you for pointing this out. Table 2 has been corrected by adding commas.
The literature n-hexadecane results are part of our discussion of LEV and LEV/XYL substrate films, and we now include it. We change the sentence on line 363
"Also, γ of solid LEV and LEV/XYL is similar to NO$_3$ uptake by solid alkane substrates and monolayers (Knopf et al., 2011;Gross and Bertram, 2009;Moise et al., 2002)."
to
 "Also, γ of solid LEV and LEV/XYL is similar to NO$_3$ uptake by solid alkane substrates and monolayers, e.g. n-hexadecane given in Table 2 (Knopf et al., 2006;Knopf et al., 2011;Gross and Bertram, 2009;Moise et al., 2002)."